# DOME: Distributed Online Learning based Multi-Estimate Fusion for Cooperative Predictive Target Tracking Using a Robotic Swarm

**Shubhankar Gupta**                                        *shubhankarg@iisc.ac.in*
*Artificial Intelligence and Robotics Lab*
*Department of Aerospace Engineering*
*Indian Institute of Science, Bengaluru, India*

**Saksham Sharma**                                         *saksham.prob@gmail.com*
*Artificial Intelligence and Robotics Lab*
*Department of Aerospace Engineering*
*Indian Institute of Science, Bengaluru, India*

**Suresh Sundaram**                                         *vssuresh@iisc.ac.in*
*Artificial Intelligence and Robotics Lab*
*Department of Aerospace Engineering*
*Indian Institute of Science, Bengaluru, India*

**Reviewed on OpenReview:** *https://openreview.net/forum?id=aF5PHD6vll*

## Abstract

This paper investigates cooperative predictive target tracking using a robotic swarm operating under high prediction bias and communication uncertainty. The robots interact over a randomly time-varying communication network and exhibit heterogeneity in onboard sensors and prediction algorithms. To address these challenges, a Distributed Online learning-based Multi-Estimate (DOME) fusion algorithm is proposed, which performs a collaborative weighted fusion of local and socially shared predictions. The fusion weights are adapted online using feedback from a prediction loss. Theoretical analysis establishes that conditional expectations of the fusion weights converge under reasonable assumptions. Simulation studies demonstrate that DOME outperforms both covariance-based and online learning-based decentralized fusion baselines, achieving 74% and 72.4% lower prediction loss in performance and scalability tests, respectively – particularly under conditions involving significant model drift and communication unreliability. Further, DOME fusion is implemented in a ROS-Gazebo simulation environment.

## 1 Introduction

With robotics technologies becoming more economical, smaller, and reliable, robotic swarms are being considered attractive for use in hazardous and uncertain environments (Mohiuddin et al., 2020). A robotic swarm has a wide range of applications, such as search and rescue (Scherer et al., 2015), precision drug delivery (Nelson & Pané, 2023), and surveillance (Saska et al., 2016), among others, that involve tracking a target as one of its fundamental tasks. A key challenge in target tracking with sensor-equipped robots is the look-ahead trajectory prediction (Hao et al., 2018); robots use predicted paths to plan their motion for effective tracking performance.

Most prior work on target tracking using a robotic swarm focuses on path planning and control. Approaches include coordinated control via formation flying (Ma & Hovakimyan, 2015; Sun et al., 2021) and region-based strategies (Jung & Sukhatme, 2006). Wang & Gu (2011) combined distributed Kalman filtering for target

localization with flocking control for tracking and collision avoidance. Hausman et al. (2015) presented a centralized control scheme that leverages onboard sensing for target estimation. Subbarao & Ahmed (2017) addressed tracking under dynamic network topologies using pinning control and consensus on target states (Wang & Su, 2014). In contrast, this work focuses on the estimation/prediction aspect of cooperative target tracking, rather than control or planning.

Heterogeneous sensors and prediction algorithms within a robotic swarm can be systematically exploited to mitigate uncertainty in target trajectory estimation (Rizk et al., 2019). Motivated by this capability, this paper proposes the Distributed Online learning–based Multi-Estimate (DOME) fusion algorithm, which employs an implicit, adaptive, consensus-like update mechanism to ensure robust performance under high uncertainty and adversarial conditions. The cooperative target-tracking problem involves robots connected over a random communication network, each running a potentially different prediction algorithm to generate look-ahead trajectories of the target. Prediction accuracy may vary over time due to algorithmic differences, scenario-specific optimizations (cf. no free lunch theorem; Murphy, 2012), or environmental and system uncertainties. DOME performs a collaborative, online weighted fusion of local and social predictions. The fusion weights are learned via a prediction loss-driven process inspired by the exponentially weighted forecaster framework (Cesa-Bianchi & Lugosi, 2006). This ensures that more accurate predictions receive higher weight, thereby improving tracking performance.

A convergence analysis of DOME's learning weights is presented, demonstrating that their expected values converge over time. DOME is evaluated in a simulated environment featuring random communication link failures and significant dynamic biases or drift in predictions. DOME is benchmarked against several decentralized fusion baselines, including Averaging Consensus Fusion (ACF); four covariance-based methods – Kalman Consensus Fusion (KCF), Covariance Intersection Consensus Fusion (CICF), Covariance Intersection + Covariance Union Consensus Fusion (CICUCF), and Separate Bias Kalman Consensus Fusion (SBKCF) (Friedland, 1969; Ignagni, 1990); and two decentralized online learning approaches – Greedy-Local (GL) and Distributed Mirror Descent (DMD) (Shahrampour & Jadbabaie, 2017a;b). Simulation results show that DOME consistently outperforms all baselines, achieving at least 74% lower prediction loss in performance evaluations. In terms of scalability, DOME also demonstrates a minimum of 72.4% reduction in prediction loss relative to the compared methods. Additionally, the algorithm is validated in a ROS-Gazebo simulation environment.

Although the covariance-based baselines are derived from earlier foundational methods (e.g., Kalman Filter, Covariance Intersection, and Covariance Union), they remain widely adopted in practice due to their simplicity, low computational overhead, and closed-form realizations. Recent works such as Chang et al. (2021), Daass et al. (2021), Wang et al. (2021), and Jia et al. (2023), use these methods for multi-estimate fusion. The proposed DOME algorithm is likewise lightweight and analytically tractable, yet achieves better performance without relying on the consistency or correlation assumptions commonly required by covariance-based methods.

The remainder of the paper is organized as follows: Section 2 reviews related work, Section 3 formulates the problem, Section 4 introduces the DOME fusion algorithm, and Section 5 presents its theoretical analysis. Section 6 reports the simulation results, and Section 7 concludes the paper with a summary and a broader impact statement. Nomenclature is provided in the Appendix (A.1), along with the MATLAB simulation details (A.3), including details regarding the baselines (A.3.2). Both MATLAB and ROS-Gazebo simulation videos are provided as supplementary files. Details regarding the ROS-Gazebo simulation are included in the video itself.

## 2   Related Works

In a robotic swarm, heterogeneous sensors and prediction algorithms can be leveraged to reduce uncertainty in target trajectory estimation (Rizk et al., 2019). Multi-sensor fusion in such systems commonly employs established methods such as Kalman Filter/Fusion (KF) (Maybeck, 1982; Uhlmann, 2003), Covariance Intersection (CI) (Matzka & Altendorfer, 2009; Julier & Uhlmann, 2017), and Covariance Union (CU) (Matzka & Altendorfer, 2009; Reece & Roberts, 2010). Building on CI, Carrillo-Arce et al. (2013) proposed a decentralized cooperative localization algorithm that reduces processing and communication overhead while

maintaining consistency under asynchronous communication. Assa & Janabi-Sharifi (2015) developed a non-linear KF-based fusion framework incorporating adaptive noise compensation and iterative updates for fast dynamics. Chang et al. (2021) used CI to maintain consistency and enhance resilience in multi-robot localization. Daass et al. (2021) compared KF- and CI-based architectures, identifying the partially distributed approach as optimal in terms of stability and efficiency. Wang et al. (2021) introduced a fully decentralized CU-based localization method that mitigates the effects of spurious sensor data. Jia et al. (2023) utilized CI to estimate the covariance of initialized UWB anchor positions, thereby enabling consistent data fusion in their distributed visual-inertial ranging odometry framework. However, these covariance-based approaches rely on assumptions regarding consistency and inter-estimate correlations. In scenarios with significant dynamic biases or drift, particularly when bias and covariance are uncorrelated, such methods may degrade or fail.

Unlike covariance-based methods, distributed online learning frameworks (Cesa-Bianchi & Lugosi, 2006; Chang et al., 2020) operate without requiring covariance information or assumptions about consistency and correlation. While most existing works focus on general algorithmic settings (Chang et al., 2020), their application and analysis within robotics remain underexplored. Shahrampour & Jadbabaie (2017a) introduced a mirror descent-based algorithm for tracking the minimizer of a time-varying convex function under adversarial noise. Similarly, Shahrampour & Jadbabaie (2017b) proposed a decentralized variant for multi-agent target tracking with unknown, unstructured disturbances. Eshraghi & Liang (2020) addressed heterogeneous networks with time-varying topology, proposing any-batch mirror descent to limit latency from slower nodes. Jiang et al. (2021) developed an asynchronous gradient-push method using asymmetric gossip and instantaneous model averaging. Eshraghi & Liang (2022) extended mirror descent with multi-iteration averaging over both decisions and gradients to improve tracking of dynamic global minimizers over time-varying networks. These algorithms typically incorporate consensus averaging with fixed, pre-defined, or equal weights. However, the lack of adaptive weighting limits their robustness in challenging conditions involving large or time-varying biases, drift, or communication uncertainties.

## 3 Problem Formulation

The decentralized cooperative target-tracking scenario (Fig. 1) involves multiple robots predicting the trajectory of a target with unknown dynamics over a random communication network. Each robot is equipped with a sensor suite and a local prediction algorithm that relies solely on its own observations. While the target is observable to all robots, differences in sensor types and algorithms may lead to varying prediction accuracies across segments of the trajectory, particularly under system or environmental uncertainties.

In practice, full observability of the target arises mainly in two situations. The first is when robots possess high-grade, long-range, wide–field-of-view sensors capable of independently tracking the target. The second occurs when centralized sensing assets, such as ground radar networks, AWACS, or satellites, monitor the target and broadcast its position to the swarm via one-way communication. In swarm robotics, this centralized sensing paradigm enhances situational awareness, reduces individual sensing demands, and enables scalable, coordinated operation over large environments. Check Appendix A.4 for more details.

Robots that share a direct communication link are referred to as neighbors. The topology of the random communication network is represented by a bi-directional random graph $G(t)$, where $t$ is the discrete-time variable, whose underlying base graph topology is denoted as $\bar{G}$; the links (edges) in the digraph $\bar{G}$ drop with a probability of $p_{ld}$, called as the link-drop probability, thus, representing communication failure. Robots can exchange information only with their neighbors and have no knowledge of the global network structure. Let $\Omega_{t,i}$ denote the set of neighbors of the $i^{th}$ robot at time $t$, as defined by the communication graph $G(t)$. Define $\Lambda_{t,i} := \Omega_{t,i} \cup \{i\}$, i.e., the set of $i^{th}$ robot's neighbors including itself. Similarly, let $\bar{\Omega}_i$ denote the set of neighbors of the $i^{th}$ robot as per the base-graph topology $\bar{G}$, and define $\bar{\Lambda}_i := \bar{\Omega}_i \cup \{i\}$.

Let $N$ denote the number of robots in the swarm, with each robot indexed by $i \in [N]$. The $i^{th}$ robot runs a prediction algorithm $A_i$ that estimates the target's look-ahead trajectory using real-time onboard sensor data. The set $\{A_i\}_{i=1}^N$ may consist of non-identical algorithms – either differing in type or parameterization – due to heterogeneous or complementary sensors and models. Consequently, prediction accuracy may vary

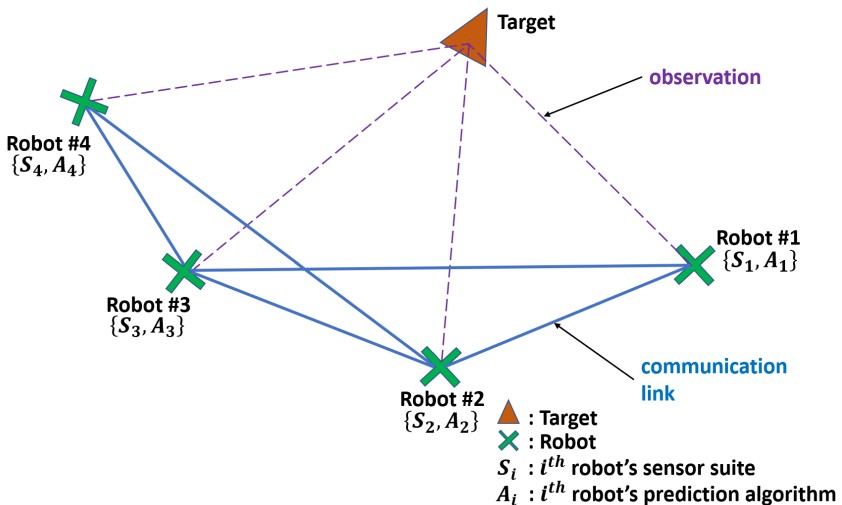

Figure 1: Decentralized cooperative predictive target-tracking using a robotic swarm. Each robot $i \in [N]$ is equipped with a local prediction algorithm $A_i$. The robots share information over the communication network and aim to cooperatively predict the target position.

across robots and along different segments of the target's trajectory. Even identical algorithms can yield varying accuracy due to system or environmental uncertainties (e.g., hardware or software failures).

**Robot Model:** consider the following discrete time 3-DOF kinematic model for the $i^{th}$ robot, where $\Delta T$ is the sampling period (seconds), $\forall i \in [N]$

$$\mathbf{x}_{t+1,i} = \mathbf{x}_{t,i} + \Delta T \begin{bmatrix} \cos \phi_{t,i} & -\sin \phi_{t,i} \\ \sin \phi_{t,i} & \cos \phi_{t,i} \end{bmatrix} \bar{\mathbf{v}}_{t,i} \tag{1a}$$

$$\phi_{t+1,i} = \phi_{t,i} + \Delta T \bar{w}_{t,i} \tag{1b}$$

where $\mathbf{x}_{t,i} \in \mathbb{R}^2$ is the 2-D position vector (in $m$), $\bar{\mathbf{v}}_{t,i} \in \mathbb{R}^2$ is the body-axis velocity vector ($m/s$), $\phi_{t,i} \in \mathbb{R}$ is the heading angle (radians), and $\bar{w}_{t,i} \in \mathbb{R}$ is the yaw rate ($rad/s$) of the $i^{th}$ robot at discrete-time $t$, respectively. Here, the body-axis velocity $\bar{\mathbf{v}}_{t,i}$ and yaw rate $\bar{w}_{t,i}$ are the bounded control inputs for the $i^{th}$ robot.

**Target Model:** the set of equations (1) also serves as the target's kinematic model, but the target's dynamic model is unknown. The target's position vector $\mathbf{x}_{t,B} \in \mathbb{R}^2$ (in $m$), heading angle $\phi_{t,B} \in \mathbb{R}$ (radians), body-axis velocity $\bar{\mathbf{v}}_{t,B} \in \mathbb{R}^2$ (m/s), and yaw rate $\bar{w}_{t,B} \in \mathbb{R}$ ($rad/s$), respectively, can be represented by replacing $i$ with $B$ (Bogey) in the set of equations (1). $\bar{\mathbf{v}}_{t,B}$ and $\bar{w}_{t,B}$ are the bounded control inputs for the target at time $t$, which are considered unknown to the robots because the target's dynamics is unknown.

*Remark* 1. While MATLAB simulations (Section 6) assume an omnidirectional (holonomic) kinematic model for both robots and the target, the proposed DOME fusion framework is model-agnostic. ROS-Gazebo simulations (check supplementary video) demonstrate its applicability to non-holonomic systems.

**Control Law:** for the $i^{th}$ robot, the translational control law consists of two terms as given below

$$\bar{\mathbf{v}}_{t,i} = \bar{\mathbf{v}}_{t,i}^R + \Delta \bar{\mathbf{v}}_{t,i} \tag{2}$$

where $\bar{\mathbf{v}}_{t,i}^R$ is the $i^{th}$ robot's reference command signal responsible for chasing the target by using its $\tau$-step look-ahead estimate of the target's trajectory, and $\Delta \bar{\mathbf{v}}_{t,i}$ is the $i^{th}$ robot's correction control signal responsible for avoiding collisions with other robots. Furthermore, consider a heading angle requirement such that the robots are required to yaw in a way that their heading direction points towards their 1-step look-ahead estimate of the target's trajectory. A detailed description of the control law used in the simulations (Section 6) is provided in the Appendix (A.3.1).

*Remark* 2. As the focus of this work is on target trajectory estimation/prediction, a simplified control strategy is employed. However, the proposed DOME fusion framework is compatible with more advanced control schemes (e.g., obstacle avoidance (Sun et al., 2014) and formation control (Cheng et al., 2005)), depending on application needs.

**Abstract Model for Prediction Algorithm** $A_i$**:** let $\hat{\mathbf{x}}_{(t+\tau|t),B}^{A_i} \in \mathbb{R}^2$ denote the $\tau$-step look-ahead prediction of the target's position by algorithm $A_i$ at time $t$. We model this prediction as:

$$\hat{\mathbf{x}}_{(t+\tau|t),B}^{A_i} = \mathbf{x}_{t+\tau,B} + \zeta_{t,i}^{\tau} \tag{3}$$

where $\mathbf{x}_{t+\tau,B}$ is the true target position at time $t+\tau$, and $\zeta_{t,i}^{\tau} \in \mathbb{R}^2$ is the prediction drift at time $t$. The drift $\zeta_{t,i}^{\tau}$ models prediction inaccuracies specific to $A_i$ and is assumed to follow an arbitrary, unknown structure – potentially non-Gaussian and without a known distribution – capturing variability in accuracy across algorithms $\{A_j\}_{j \in [N] \setminus \{i\}}$.

With the assistance of its prediction algorithm $A_i$ and its neighboring robots as per the communication network $G(t)$ at time $t$, each robot aims to estimate/predict the target's position at time $t+\tau$, i.e., $\mathbf{x}_{t+\tau,B}$.

# 4  Distributed Online Learning based Multi-Estimate Fusion

## 4.1  Conceptual Overview

The Distributed Online Learning–based Multi-Estimate (DOME) fusion framework tackles the problem of trust management in heterogeneous swarms, where robots differ in sensing and prediction quality and communication is unreliable. DOME learns which agents provide reliable predictive-tracking information and weights their contributions accordingly.

Each iteration consists of four phases (see Appendix, Fig. 10): learning, local fusion, communication, and social fusion. During the learning phase, each robot evaluates its own and neighbors' previous predictions against the current target observation and updates trust weights at two levels. **Local trust** balances the robot's onboard prediction with its previous social prediction, while **social trust** balances its local prediction against neighbors' local predictions.

In the local fusion phase, the robot combines its fresh onboard prediction with its previous social prediction using updated local trust weights to produce a robust local prediction. During the communication phase, robots share their local predictions along with associated self-trust weights with available neighbors. In the social fusion phase, each robot computes a trust-weighted average of received local predictions to generate the final social estimate for control or navigation.

To handle non-stationary environments (e.g., abrupt target maneuvers or sensor occlusions), DOME incorporates a **periodic reset** mechanism in its learning phase. Purely cumulative online learning can lead to weight saturation, where historically reliable robots retain excessive influence even after performance degrades. To mitigate this historical inertia, DOME resets trust weights to a neutral baseline every ($T_o$) steps, forcing re-evaluation based on recent data and maintaining adaptability to dynamic conditions.

## 4.2  DOME Fusion Algorithm

**Learning Phase:** at the current time step $t$, robots observe the current position of the target, $\mathbf{x}_{t,B}$. The robots use this observation as the ground truth to learn the DOME fusion weights.

The DOME fusion weights $\alpha_i(t)$ and $w_{ij}(t)$ are updated using a bounded loss function $l(\mathbf{x}, \mathbf{y}) \in [0, 1]$, with its arguments $\mathbf{x} \in \mathbb{R}^2$ and $\mathbf{y} \in \mathbb{R}^2$.

*Remark* 3. The proposed algorithm applies to any suitable set $\mathcal{X}$ with $\mathbf{x}, \mathbf{y} \in \mathcal{X}$ for which the loss function $l(\mathbf{x}, \mathbf{y}) \in [0, 1]$ is defined. While DOME fusion extends naturally to 3-D ($\mathcal{X} = \mathbb{R}^3$) or n-D ($\mathcal{X} = \mathbb{R}^n$) settings, a 2-D case ($\mathcal{X} = \mathbb{R}^2$) is used here for simplicity.

Consider the current time step neighbor set $\Omega_{t,i}$, and $\Lambda_{t,i} = \Omega_{t,i} \cup \{i\}$ as the set of $i^{th}$ robot's neighbors including itself at time $t$.

Denote $\hat{\mathbf{x}}^i_{(t|t-1),B}$ as the $i^{th}$ robot's 1-step look-ahead social prediction of the target's position at the previous time step $t-1$, calculated at time $t-1$ using the previous time weights $w_{ij}(t-1)$ and the previous time 1-step look-ahead local predictions $\hat{\mathbf{x}}^{L_j}_{(t|t-1),B}$, where $j \in \Lambda_{t-1,i}$, can be given as follows:

$$\hat{\mathbf{x}}^i_{(t|t-1),B} = \sum_{\forall j \in \Lambda_{t-1,i}} w_{ij}(t-1)\,\hat{\mathbf{x}}^{L_j}_{(t|t-1),B} \tag{4}$$

where the $i^{th}$ robot's previous time 1-step look-ahead local predictions $\hat{\mathbf{x}}^{L_i}_{(t|t-1),B}$, calculated at time $t-1$ using algorithm $A_i$'s previous time 1-step look-ahead prediction $\hat{\mathbf{x}}^{A_i}_{(t|t-1),B}$, 1-step look-ahead social prediction $\hat{\mathbf{x}}^{A_i}_{(t|t-1),B}$ at time $t-2$, and the previous time weights $\alpha_i(t-1)$, $\forall i \in [N]$, can be given as follows:

$$\hat{\mathbf{x}}^{L_i}_{(t|t-1),B} = \alpha_i(t-1)\,\hat{\mathbf{x}}^{A_i}_{(t|t-1),B} + (1-\alpha_i(t-1))\,\hat{\mathbf{x}}^i_{(t-1|t-2),B} \tag{5}$$

The DOME fusion's local weights $\alpha_i(t)$ are updated as follows:

$$\hat{\alpha}_i(t) = \hat{\alpha}_i(t-1)\exp\left(-\eta_\alpha\,l(\mathbf{x}_{t,B}, \hat{\mathbf{x}}^{A_i}_{(t|t-1),B})\right) \tag{6a}$$

$$\hat{\alpha}'_i(t) = \hat{\alpha}'_i(t-1)\exp\left(-\eta_\alpha\,l(\mathbf{x}_{t,B}, \hat{\mathbf{x}}^i_{(t-1|t-2),B})\right) \tag{6b}$$

$$\alpha_i(t) = \frac{\hat{\alpha}_i(t)}{\hat{\alpha}_i(t) + \hat{\alpha}'_i(t)} \tag{6c}$$

where $\eta_\alpha > 0$ is the local weights' learning rate. The weights are initialized as $\hat{\alpha}_i(0) = 1$ and $\hat{\alpha}'_i(0) = 1$. Note that $\alpha_i(t) \in [0,1]$.

The DOME fusion's social weights $w_{ij}(t)$ are updated as follows:

$$\hat{w}_{ii}(t) = \hat{w}_{ii}(t-1)\exp\left(-\eta_w\,l(\mathbf{x}_{t,B}, \hat{\mathbf{x}}^{L_i}_{(t|t-1),B})\right) \tag{7a}$$

$$w_{ij}(t) = \mathbf{1}(j \in \Lambda_{t,i})\frac{\hat{w}_{jj}(t)}{\sum_{j' \in \Lambda_{t,i}} \hat{w}_{j'j'}(t)} \tag{7b}$$

where $\eta_w > 0$ is the social weights' learning rate, and $\mathbf{1}(\cdot)$ is the indicator function; $\mathbf{1}(j \in \Lambda_{t,i}) = 1$ if the condition $j \in \Lambda_{t,i}$ is satisfied, otherwise $\mathbf{1}(j \in \Lambda_{t,i}) = 0$. The weights are initialized as $\hat{w}_{ii}(0) = 1$. Note that $w_{ij}(t) \in [0,1]$ and $\sum_{j \in \Lambda_{t,i}} w_{ij}(t) = 1$.

DOME fusion weights $\alpha_i(t)$ and $w_{ij}(t)$ are reset every $T_o$ time steps to 1, mitigating bias accumulation under high system and environmental uncertainty. Furthermore, a decentralized normalization scheme is employed to normalize the weights after each update (see Appendix A.2 for more details).

*Remark* 4. Note that the learning phase assumes access to the true current target position $\mathbf{x}_{t,B}$, which is required to compute the loss used for updating the DOME fusion weights. In practical deployments, this assumption can be approximated through a hierarchical perception architecture, in which a high-precision perception module or an external sensing asset provides an accurate, current-time reference signal that serves as ground truth for updating the DOME weights. For example, modern LiDAR–Inertial Odometry methods such as LIO-SAM (Shan et al., 2020) and FAST-LIO2 (Xu et al., 2022) achieve centimeter-level accuracy on benchmarks; their estimates can serve as proxy ground truth to compute loss and update fusion weights. Additionally, Yan et al. (2022) show that networked radar resources can provide improved fused target states to distributed agents; such broadcasts can be treated as the true current target position $\mathbf{x}_{t,B}$ to update the DOME fusion weights. Check Appendix A.5 for a detailed theoretical and empirical analysis of the impact of noisy observations on the DOME weight updates.

**Local Prediction Fusion Phase:** using the learned weights $\alpha_i(t)$, the $i^{th}$ robot's $\tau$-step look-ahead local prediction of the target's position at time $t$. i.e., $\hat{\mathbf{x}}^{L_i}_{(t|t-1),B}$, is given as:

$$\hat{\mathbf{x}}^{L_i}_{(t+\tau|t),B} = \alpha_i(t)\,\hat{\mathbf{x}}^{A_i}_{(t+\tau|t),B} + (1-\alpha_i(t))\,\hat{\mathbf{x}}^i_{(t+\tau-1|t-1),B} \tag{8}$$

where $\hat{\mathbf{x}}^i_{(t+\tau-1|t-1),B}$ is the $i^{th}$ robot's previous time $\tau$-step look-ahead social prediction of the target's position.

**Communication Phase:** the $i^{th}$ robot shares the information $\{t, i, \hat{\mathbf{x}}^{L_i}_{(t+1|t),B}, \hat{\mathbf{x}}^{L_i}_{(t+\tau|t),B}, \hat{w}_{ii}(t)\}$ with its neighbors $j \in \Omega_{t,i}$, and receives $\{t, j, \hat{\mathbf{x}}^{L_j}_{(t+1|t),B}, \hat{\mathbf{x}}^{L_j}_{(t+\tau|t),B}, \hat{w}_{jj}(t)\}$ from its neighbors $j \in \Omega_{t,i}$.

**Social Prediction Fusion Phase:** using equation (7b) along with the received $\hat{w}_{jj}(t)$ weights via communication, we get the learned weights $w_{ij}(t)$. Further, using the local predictions received from the neighbors, the $i^{th}$ robot's $\tau$-step look-ahead social prediction at time $t$ is calculated as follows:

$$\hat{\mathbf{x}}^i_{(t+\tau|t),B} = \sum_{\forall j \in \Lambda_{t,i}} w_{ij}(t)\,\hat{\mathbf{x}}^{L_j}_{(t+\tau|t),B} \tag{9}$$

Note that the $i^{th}$ robot's $\tau$-step look-ahead social prediction $\hat{\mathbf{x}}^i_{(t+\tau|t),B}$ is considered as the final $\tau$-step look-ahead estimate of the target's position, and thus, is used by the control law for calculating velocity reference (equation (2)) and yaw-rate commands (A.3.1).

*Remark* 5. Note that the learning phase involves comparing previous time 1-step look-ahead predictions with the current time position of the target (using prediction loss) to update the weights. The learned weights are then used to get the current time $\tau$-step look-ahead predictions of the target's position.

DOME fusion is summarized in Algorithm 1.

---

**Algorithm 1** DOME (for the $i^{th}$ robot, $i \in [N]$)

---

    **Choose:** $T, T_o, \tau \geq 1$ (integers); $\eta_\alpha, \eta_w > 0$
    **Initialize:** $\hat{w}_{ii}(0) = 1$, $\hat{\alpha}_i(0) = 1$, $\hat{\alpha}'_i(0) = 1$,
       $\hat{\mathbf{x}}^i_{(0|-1),B} = \hat{\mathbf{x}}^{A_i}_{(1|0),B}$, $\hat{\mathbf{x}}^i_{(\tau-1|-1),B} = \hat{\mathbf{x}}^{A_i}_{(\tau|0),B}$
    **Iteration** at discrete time step $t = 0, 1, 2, \cdots$ :

1: Observe $\mathbf{x}_{t,B}$
2: **if** $t > 0$ **then**
3:     $\hat{\alpha}_i(t) = \hat{\alpha}_i(t-1) \exp\left(-\eta_\alpha l(\hat{\mathbf{x}}^{A_i}_{(t|t-1),B}, \mathbf{x}_{t,B})\right)$
4:     $\hat{\alpha}'_i(t) = \hat{\alpha}'_i(t-1) \exp\left(-\eta_\alpha l(\hat{\mathbf{x}}^i_{(t-1|t-2),B}, \mathbf{x}_{t,B})\right)$
5:     $\hat{w}_{ii}(t) = \hat{w}_{ii}(t-1) \exp\left(-\eta_w l(\hat{\mathbf{x}}^{L_i}_{(t|t-1),B}, \mathbf{x}_{t,B})\right)$
6: **end if**
7: *Periodic Reset*: re-initialize the weights $\hat{\alpha}_i(t)$, $\hat{\alpha}'_i(t)$, and $\hat{w}_{ii}(t)$ to 1 after every $T_o$ discrete time steps
8: $\alpha_i(t) = \frac{\hat{\alpha}_i(t)}{\hat{\alpha}_i(t)+\hat{\alpha}'_i(t)}$
9: $\hat{\mathbf{x}}^{L_i}_{(t+1|t),B} = \alpha_i(t)\hat{\mathbf{x}}^{A_i}_{(t+1|t),B} + (1-\alpha_i(t))\hat{\mathbf{x}}^i_{(t|t-1),B}$
10: $\hat{\mathbf{x}}^{L_i}_{(t+\tau|t),B} = \alpha_i(t)\hat{\mathbf{x}}^{A_i}_{(t+\tau|t),B} + (1-\alpha_i(t))\hat{\mathbf{x}}^i_{(t-1+\tau|t-1),B}$
11: transmit $\{t, i, \hat{\mathbf{x}}^{L_i}_{(t+1|t),B}, \hat{\mathbf{x}}^{L_i}_{(t+\tau|t),B}, \hat{w}_{ii}(t)\}$ and receive $\{t, j, \hat{\mathbf{x}}^{L_j}_{(t+1|t),B}, \hat{\mathbf{x}}^{L_j}_{(t+\tau|t),B}, \hat{w}_{jj}(t)\}$ from the communicating neighboring robots, $\forall j \in \Omega_{t,i}$
12: $\Lambda_{t,i} = \Omega_{t,i} \cup \{i\}$
13: $w_{ij}(t) = \mathbf{1}(j \in \Lambda_{t,i})\frac{\hat{w}_{jj}(t)}{\sum_{j' \in \Lambda_{t,i}} \hat{w}_{j'j'}(t)}$
14: $\hat{\mathbf{x}}^i_{(t+1|t),B} = \sum_{\forall j \in \Lambda_{t,i}} w_{ij}(t)\hat{\mathbf{x}}^{L_j}_{(t+1|t),B}$
15: $\hat{\mathbf{x}}^i_{(t+\tau|t),B} = \sum_{\forall j \in \Lambda_{t,i}} w_{ij}(t)\hat{\mathbf{x}}^{L_j}_{(t+\tau|t),B}$

---

## 5 Convergence Analysis

This section presents the theoretical convergence analysis of DOME fusion weights. Only the analysis of the social weights $w_{ij}(t)$ is presented; the analysis for local weights $\alpha_i(t)$ follows a similar procedure.

Without loss of generality, the DOME weights $w_{ij}(t)$ are analyzed without considering periodic resets to examine their convergence behavior immediately after and just before a reset. Each reset instant can be

treated as $t = 0$, making the analysis applicable to any interval between two successive resets. In practice, with a periodic reset interval $T_o$, a higher learning rate enables faster convergence before the next reset, while a lower rate may prevent full convergence. Nonetheless, the underlying convergence behavior remains consistent with the theoretical analysis presented.

Denote $l_{t,i}^A := l(\mathbf{x}_{t,B}, \hat{\mathbf{x}}_{(t|t-1),B}^{A_i})$, and $L_{t,i}^A = \sum_{s=1}^{t} l_{s,i}^A$. Denote $l_{t,i}^S := l(\mathbf{x}_{t,B}, \hat{\mathbf{x}}_{(t|t-1),B}^i)$, and $L_{t,i}^S = \sum_{s=1}^{t} l_{s,i}^S$. Denote $l_{t,i}^L := l(\mathbf{x}_{t,B}, \hat{\mathbf{x}}_{(t|t-1),B}^{L_i})$, and $L_{t,i}^L = \sum_{s=1}^{t} l_{s,i}^L$.

Note that the loss function $l(\mathbf{x}, \mathbf{y}) \in [0, 1]$ is bounded and its arguments $\mathbf{x} \in \mathbb{R}^2$ and $\mathbf{y} \in \mathbb{R}^2$.

The history $\mathcal{H}_{t,i}$ relevant to the $i^{th}$ robot's learning phase just before the communication phase occurs at time $t$, is defined as follows:

$$\mathcal{H}_{t,i} := (\{\Omega_{s,i}\}_{s=1}^{t-1}, \{\hat{\mathbf{x}}_{(s+1|s),B}^{A_i}\}_{s=1}^{t}, \{\mathbf{x}_{s,B}\}_{s=1}^{t}, \{\{\hat{\mathbf{x}}_{(s+1|s),B}^{L_i}\}_{\forall j \in \bar{\Omega}_i}\}_{s=1}^{t}) \tag{10}$$

where $\bar{\Omega}_i$ is the neighbor set of the $i^{th}$ robot as per the base-graph topology $\bar{G}$ of the random communication network $G(t)$, whereas $\Omega_{s,i}$ is the neighbor set of the $i^{th}$ robot as per the random communication network $G(s)$ at time $s$.

Let $n_i = |\bar{\Omega}_i|$ denote the maximum number of neighbors the $i^{th}$ robot can have in the base graph $\bar{G}$, where $|\bar{\Omega}_i|$ denotes the cardinality of the set $\bar{\Omega}_i$. The time-varying communication graph $G(t)$ is an undirected random subgraph of $\bar{G}$, with each link independently dropping with probability $p_{ld}$. At time $t$, robot $i$ can communicate with any subset of its $n_i$ neighbors; the number of possible neighbor sets of size $k$ is $^{n_i}C_k$, for $k = 0, 1, \ldots, n_i$. Let $\Omega_i^{k,l}$ denote the $l^{\text{th}}$ such subset with $|\Omega_i^{k,l}| = k$. Further, define $\Lambda_i^{k,l} := \Omega_i^{k,l} \cup \{i\}$. Then $\Omega_{t,i} \in \{\Omega_i^{k,l}\}_{k=0:n_i, \; l=1:^{n_i}C_k}$, and $\Omega_{t,i} \subseteq \bar{\Omega}_i$. Here, $k = 0 : n_i$ implies $k = 0, 1, \cdots, n_i$ and $l = 1 : ^{n_i}C_k$ implies $l = 1, \cdots, ^{n_i}C_k$.

Let $N_i(t) = |\Omega_{t,i}|$ denote the number of neighbors the $i^{th}$ robot has at time $t$. Since each link in the base graph $\bar{G}$ drops independently with probability $p_{ld}$, $N_i(t)$ follows a Binomial distribution: $N_i(t) \sim B(n_i, p_{ld})$, where $n_i = |\bar{\Omega}_i|$.

For some $i \in [N]$, the conditional expectation of $N_i(t)$ can be written as follows:

$$\mathbb{E}[N_i(t)|i] = \sum_{k=0}^{n_i} {}^{n_i}C_k (1 - p_{ld})^k p_{ld}^{n_i - k} k = (1 - p_{ld}) n_i \tag{11}$$

where $\mathbb{E}[\cdot]$ is the expectation operator. Note that $\mathbb{E}[N_i(t)|i]$ is independent of $t$. Further, note that

$$\sum_{k=0}^{n_i} \sum_{l=1}^{^{n_i}C_k} (1 - p_{ld})^k p_{ld}^{n_i - k} = \sum_{k=0}^{n_i} {}^{n_i}C_k (1 - p_{ld})^k p_{ld}^{n_i - k} = 1 \tag{12}$$

Consider $0 < \hat{w}_{ii}(0) \le 1$, where $i \in [N]$. For $t = 1, 2, \cdots, T$, based on equation (7a), we get $\hat{w}_{jj}(t) = \hat{w}_{jj}(0) \exp\left(-\eta_w L_{t,j}^L\right), \forall j \in [N]$.

Consider the weight $w_{ij}(t)$. After some mathematical manipulation, the expectation of $w_{ij}(t)$ conditioned on the history $\mathcal{H}_{t,i}, \forall j \in \bar{\Lambda}_i$, can be written as follows:

$$\mathbb{E}[w_{ij}(t)|\mathcal{H}_{t,i}] = \sum_{k=0}^{n_i} \sum_{l=1}^{^{n_i}C_k} (1 - p_{ld})^k p_{ld}^{n_i - k} w_{ij}^{k,l}(t) \tag{13}$$

where

$$w_{ij}^{k,l}(t) := \mathbf{1}(j \in \Lambda_i^{k,l}) \frac{\hat{w}_{jj}(t)}{\sum_{q \in \Lambda_i^{k,l}} \hat{w}_{qq}(t)} \tag{14}$$

Here, $\mathbf{1}(\cdot)$ is the indicator function; $\mathbf{1}(j \in \Lambda_i^{k,l}) = 1$ if the condition $j \in \Lambda_i^{k,l}$ is satisfied, otherwise $\mathbf{1}(j \in \Lambda_i^{k,l}) = 0$.

Define $j_{*,i}^{k,l}(t) := \arg\min_{j' \in \Lambda_i^{k,l}} L_{t,j'}^L$, i.e., $j_{*,i}^{k,l}(t)$ is the index of the robot which incurs the least cumulative loss among all other robots in the index set $\Lambda_i^{k,l} = \Omega_i^{k,l} \cup \{i\}$ at time $t$, where $\Omega_i^{k,l}$ is one of the many possible neighbor sets of the $i^{th}$ robot at time $t$, such that $|\Omega_t^{k,l}| = k$, where $k = 0, 1, \cdots, n_i$, and $l = 1, 2, \cdots, {}^{n_i}C_k$.

**Assumption 1.** *For each $k \in \{0, 1, \cdots, n_i\}$ and $l \in \{1, 2, \cdots, {}^{n_i}C_k\}$ pair, $\lim_{t\to\infty} j_{*,i}^{k,l}(t)$ exists uniquely, such that $\lim_{t\to\infty} j_{*,i}^{k,l}(t) = j_{\infty,i}^{k,l}$.*

*Remark* 6. Assumption 1 implies that the performance configuration (in terms of prediction loss) gets fixed as $t \to \infty$, i.e., for the $i^{th}$ robot at $t \to \infty$, there is a unique robot (either itself or its neighbor given by the index $j_{\infty,i}^{k,l}$) that incurs the least cumulative loss out of all the robots in the set $\Lambda_i^{k,l}$.

Note that the cumulative loss satisfies $0 \leq L_{t,j}^L \leq t$ (since the loss function $l(\cdot,\cdot) \in [0,1]$), $\forall j \in \Lambda_i^{k,l}$, and $L_{t,j_{*,i}^{k,l}(t)}^L < L_{t,j}^L$ (due to $j_{*,i}^{k,l}(t)$'s definition), $\forall j \in \Lambda_i^{k,l} \setminus \{j_{*,i}^{k,l}(t)\}$.

**Assumption 2.** $L_{t,j}^L - L_{t,j_{*,i}^{k,l}(t)}^L \geq \epsilon t^\beta > 0$, *such that $\beta \in (0,1]$ and $\epsilon \in (0,1]$, $\forall j \in \Lambda_i^{k,l} \setminus \{j_{*,i}^{k,l}(t)\}$.*

*Remark* 7. Assumption 2 implies that the cumulative loss difference between robot $j$ and the best-performing robot $j_{*,i}^{k,l}(t)$ grows at most linearly ($\beta = 1$) or sub-linearly ($0 < \beta < 1$) with time $t$, with both the rate and magnitude finite but arbitrarily small ($0 < \epsilon \ll 1$, $0 < \beta \ll 1$). In practice, this holds when the best robot $j_{*,i}^{k,l}(t)$ remains fixed for finite durations and may change intermittently.

**Lemma 1.** *Under assumptions 1 and 2, for $k = 0, 1, \cdots, n_i$, and $l = 1, 2, \cdots, {}^{n_i}C_k$, the weights $w_{ij}^{k,l}(t)$ satisfy the following:*

$$\lim_{t\to\infty} w_{ij}^{k,l}(t) = 0, \quad \forall j \in \Lambda_i^{k,l} \setminus \{j_{\infty,i}^{k,l}\} \tag{15}$$

*and*

$$\lim_{t\to\infty} w_{ij_{*,i}^{k,l}(t)}^{k,l}(t) = \lim_{t\to\infty} w_{ij_{\infty,i}^{k,l}}^{k,l}(t) = 1 \tag{16}$$

*where $j_{*,i}^{k,l}(t)$ is the index of the neighbor of the $i^{th}$ robot whose individual prediction incurs the least cumulative loss among all other robots in the index set $\Lambda_i^{k,l} = \Omega_i^{k,l} \cup \{i\}$ at time $t$, i.e., $j_{*,i}^{k,l}(t) = \arg\min_{q \in \Lambda_i^{k,l}} L_{t,q}^L$, $\forall i \in [N]$.*

*Proof.* For some $k \in \{0, 1, \cdots, n_i\}$ and $l \in \{1, 2, \cdots, {}^{n_i}C_k\}$, consider the weight $w_{ij}^{k,l}(t)$, which can be re-written as follows; use $\hat{w}_{jj}(t) = \hat{w}_{jj}(0) \exp\left(-\eta_w L_{t,j}^L\right)$ in eq.(14) and multiply both numerator and denominator by $\exp\left(\eta_w L_{t,j_{*,i}^{k,l}(t)}^L\right)$ to get:

$$w_{ij}^{k,l}(t) = \frac{\mathbf{1}(j \in \Lambda_i^{k,l})\hat{w}_{jj}(0) \exp\{-\eta_w(L_{t,j}^L - L_{t,j_{*,i}^{k,l}(t)}^L)\}}{\sum_{q \in \Lambda_i^{k,l}} \hat{w}_{qq}(0) \exp\{-\eta_w(L_{t,q}^L - L_{t,j_{*,i}^{k,l}(t)}^L)\}} \tag{17}$$

In the above equation, separating the $\hat{w}_{j_{*,i}^{k,l}(t)j_{*,i}^{k,l}(t)}(0)$ term from the summation in the denominator yields:

$$w_{ij}^{k,l}(t) = \frac{\mathbf{1}(j \in \Lambda_i^{k,l})\hat{w}_{jj}(0) \exp\{-\eta_w(L_{t,j}^L - L_{t,j_{*,i}^{k,l}(t)}^L)\}}{\hat{w}_{j_{*,i}^{k,l}(t)j_{*,i}^{k,l}(t)}(0) + \sum_{q \in \Lambda_i^{k,l} \setminus \{j_{*,i}^{k,l}(t)\}} \hat{w}_{qq}(0) \exp\{-\eta_w(L_{t,q}^L - L_{t,j_{*,i}^{k,l}(t)}^L)\}} \tag{18}$$

Note that for the cumulative loss, the following holds true: $0 \leq L_{t,j}^L \leq t$ (since the loss function is bounded, i.e., $l(\cdot,\cdot) \in [0,1]$), $\forall j \in \Lambda_i^{k,l}$, and $L_{t,j_{*,i}^{k,l}(t)}^L < L_{t,j}^L$ (due to $j_{*,i}^{k,l}(t)$'s definition), $\forall j \in \Lambda_i^{k,l} \setminus \{j_{*,i}^{k,l}(t)\}$. Further using assumption 2, this implies that the cumulative loss for the $j^{th}$ robot, $\forall j \in \Lambda_i^{k,l} \setminus \{j_{*,i}^{k,l}(t)\}$, satisfies

$$t \geq L_{t,j}^L - L_{t,j_{*,i}^{k,l}(t)}^L \geq \epsilon t^\beta > 0 \tag{19}$$

where $\beta \in (0,1]$ and $\epsilon \in (0,1]$, $\forall j \in \Lambda_i^{k,l} \setminus \{j_{*,i}^{k,l}(t)\}$.

Using equation (19) in equation (18), we get

$$
\frac{\mathbf{1}(j \in \Lambda_i^{k,l}) \hat{w}_{jj}(0) \exp\{-\eta_w t\}}{\hat{w}_{j_{*,i}^{k,l}(t), j_{*,i}^{k,l}(t)}(0) + \sum_{q \in \Lambda_i^{k,l} \setminus \{j_{*,i}^{k,l}(t)\}} \hat{w}_{qq}(0) \exp\{-\eta_w \epsilon t^\beta\}}
$$
$$
\leq w_{ij}^{k,l}(t) \leq
$$
$$
\frac{\mathbf{1}(j \in \Lambda_i^{k,l}) \hat{w}_{jj}(0) \exp\{-\eta_w \epsilon t^\beta\}}{\hat{w}_{j_{*,i}^{k,l}(t), j_{*,i}^{k,l}(t)}(0) + \sum_{q \in \Lambda_i^{k,l} \setminus \{j_{*,i}^{k,l}(t)\}} \hat{w}_{qq}(0) \exp\{-\eta_w t\}}
\tag{20}
$$

for $\forall j \in \Lambda_i^{k,l} \setminus \{j_{*,i}^{k,l}(t)\}$, and

$$
\frac{\hat{w}_{j_{*,i}^{k,l}(t), j_{*,i}^{k,l}(t)}(0)}{\hat{w}_{j_{*,i}^{k,l}(t), j_{*,i}^{k,l}(t)}(0) + \sum_{q \in \Lambda_i^{k,l} \setminus \{j_{*,i}^{k,l}(t)\}} \hat{w}_{qq}(0) \exp\{-\eta_w \epsilon t^\beta\}}
$$
$$
\leq w_{ij_{*,i}^{k,l}(t)}(t) \leq
$$
$$
\frac{\hat{w}_{j_{*,i}^{k,l}(t), j_{*,i}^{k,l}(t)}(0)}{\hat{w}_{j_{*,i}^{k,l}(t), j_{*,i}^{k,l}(t)}(0) + \sum_{q \in \Lambda_i^{k,l} \setminus \{j_{*,i}^{k,l}(t)\}} \hat{w}_{qq}(0) \exp\{-\eta_w t\}}
\tag{21}
$$

Now, taking $\lim_{t \to \infty}(\cdot)$ on equations (20) and (21), under assumption 1, leads to the desired result. $\qquad\square$

For each $j \in \bar{\Lambda}_i$, consider the set $M_{j,i}$ defined as follows:

$$
M_{j,i} := \{(k,l) : j = j_{\infty,i}^{k,l}; \ k \in \{0, \cdots, n_i\}, l \in \{1, \cdots, {}^{n_i}C_k\}\}
\tag{22}
$$

Further, consider the set $J_i$, which is defined as

$$
J_i := \{j : j = j_{\infty,i}^{k,l}; \ k \in \{0, \cdots, n_i\}, l \in \{1, \cdots, {}^{n_i}C_k\}, j \in \bar{\Lambda}_i\}
\tag{23}
$$

*Remark* 8. The set $M_{j,i}$ contains all the pairs $(k,l)$ for which $j = j_{\infty,i}^{k,l}$ is satisfied. The set $J_i$ consists of all the indices $j_{\infty,i}^{k,l}$, for $k = 0, 1, \cdots, n_i$, and $l = 1, 2, \cdots, {}^{n_i}C_k$.

**Theorem 1.** *Under assumptions 1 and 2 (using Lemma 1), $\forall j \in \bar{\Lambda}_i$, DOME fusion weights $w_{ij}(t)$ satisfy the following:*

$$
\lim_{t \to \infty} \mathbb{E}[w_{ij}(t) | \mathcal{H}_{t,i}] = 0, \quad \forall j \notin J_i
\tag{24}
$$

*and*

$$
\lim_{t \to \infty} \mathbb{E}[w_{ij}(t) | \mathcal{H}_{t,i}] = \sum_{\forall (k,l) \in M_{j,i}} (1 - p_{ld})^k p_{ld}^{n_i - k}, \ \forall j \in J_i
\tag{25}
$$

*where $p_{ld}$ is the communication link-drop probability.*

*Proof.* From lemma 1, $\forall (k,l) : k = 0, 1, \cdots, n_i; \ l = 1, 2, \cdots, {}^{n_i}C_k$, using the definition of the set $J_i$ as stated in equation (23), note that the following holds true:

$$
\lim_{t \to \infty} w_{ij}^{k,l}(t) = 0, \quad \forall j \notin J_i
\tag{26}
$$

and

$$
\lim_{t \to \infty} w_{ij}^{k,l}(t) = \lim_{t \to \infty} w_{ij_\infty^{k,l}}^{k,l}(t) = 1 \quad \forall j \in J_i
\tag{27}
$$

Using equations (26) and (27) in equation (13), along with the definition of the set $M_{j,i}$ as given in equation (22), leads to the desired result. $\qquad\square$

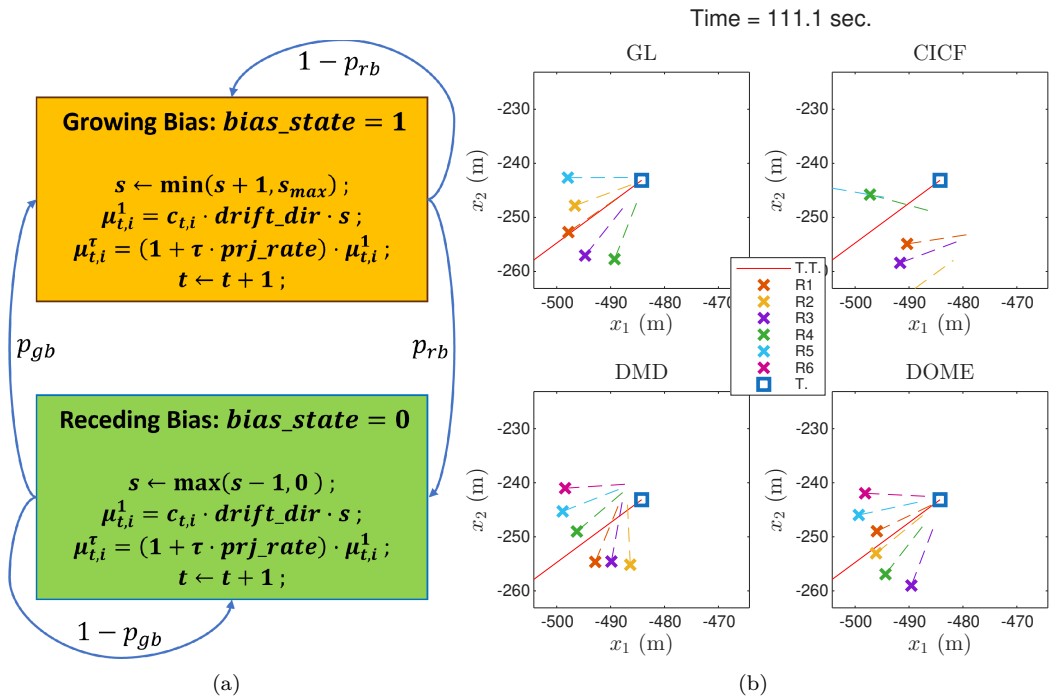

(a)                          (b)

Figure 2: (a) An event-based switching signal model for the dynamic bias term $\mu_{t,i}^{\tau}$, for $\tau > 1$. Here, $drift\_dir = [\cos\psi_{dir} \quad \sin\psi_{dir}]'$, such that $\psi_{dir}$ is sampled from $Unif.(0, 2\pi)$ at the start of every simulation run. The value for $prj\_rate$ is set to 0.05. (b) Screenshot of a simulation run in MATLAB for DOME and a few baselines; T. is the target, T.T. is the target's trajectory, and R$i$ is the $i^{th}$ robot.

## 6 Performance Evaluation

The DOME fusion algorithm is simulated for cooperative target tracking with $N = 6$ robots over a horizon of $T = 1400$ steps and a sampling period $\Delta T = 0.1$ s, using a look-ahead window of $\tau = 10$. Robots communicate over a bi-directional network with random link drops (drop probability, $p_{ld} = 0.1$), based on an undirected connected linear graph – chosen to represent the worst-case among connected topologies. Further, the loss function is set as $l(\mathbf{x}, \mathbf{y}) = \min(||\mathbf{x} - \mathbf{y}||/50, 1)$, where $|| \cdot ||$ is the Euclidean norm (2-norm) and $\mathbf{x}, \mathbf{y} \in \mathbb{R}^2$.

The drift $\zeta_{t,i}^{\tau}$ in the prediction by algorithm $A_i$ (equation (3)) is modeled as follows:

$$\zeta_{t,i}^{\tau} = \mu_{t,i}^{\tau} + \nu_{t,i}^{\tau} \tag{28}$$

where $\mu_{t,i}^{\tau} \in \mathbb{R}^2$ is a bias term (m), and $\nu_{t,i}^{\tau} \in \mathbb{R}^2$ is zero-mean Gaussian noise (m) with covariance $\mathbf{C}_{t,i}^{\tau} \in \mathbb{R}_{\geq 0}^{2 \times 2}$ (m²) at time $t$. The bias $\mu_{t,i}^{\tau}$ is modeled as an event-driven switching signal (see Fig. 2a), offering a more realistic alternative to the ramp signal used in Cho & Jiang (2012). In the *growing bias* state, the bias magnitude either increases until saturation or switches to the *receding bias* state with probability $p_{rb}$. Conversely, in the receding bias state, it decreases toward zero or switches back with probability $p_{gb}$. In Fig. 2a, the scalar $c_{t,i}$, indicative of the accuracy of algorithm $A_i$, increases with prediction error. In simulations, $p_{gb} = 0.2$ and $p_{rb} = 0.5$.

To simulate adverse system and environmental conditions, robots are randomly assigned either $c_{t,i} = 0.01$ or $c_{t,i} = 4$ at times 0.0, 23.33, 46.67, 70, and 116.67 seconds, ensuring that 3 to 4 out of $N = 6$ robots have inaccurate algorithms (i.e., $c_{t,i} = 4$). For these robots, the one-step prediction bias satisfies that each element of the vector $\mu_{t,i}^1$ lies in the range $[0, 40]$ m (see Fig. 2a, with $s_{max} = 10$). For inaccurate robots ($c_{t,i} = 4$), the associated noise covariance $\mathbf{C}_{t,i}^1$ is either $4 \cdot \text{diag}([1 \; 1])$ or $0.01 \cdot \text{diag}([1 \; 1])$, each with 50% probability. For accurate robots ($c_{t,i} = 0.01$), $\mathbf{C}_{t,i}^1 = 0.01 \cdot \text{diag}([1 \; 1])$. For $\tau > 1$, the noise grows

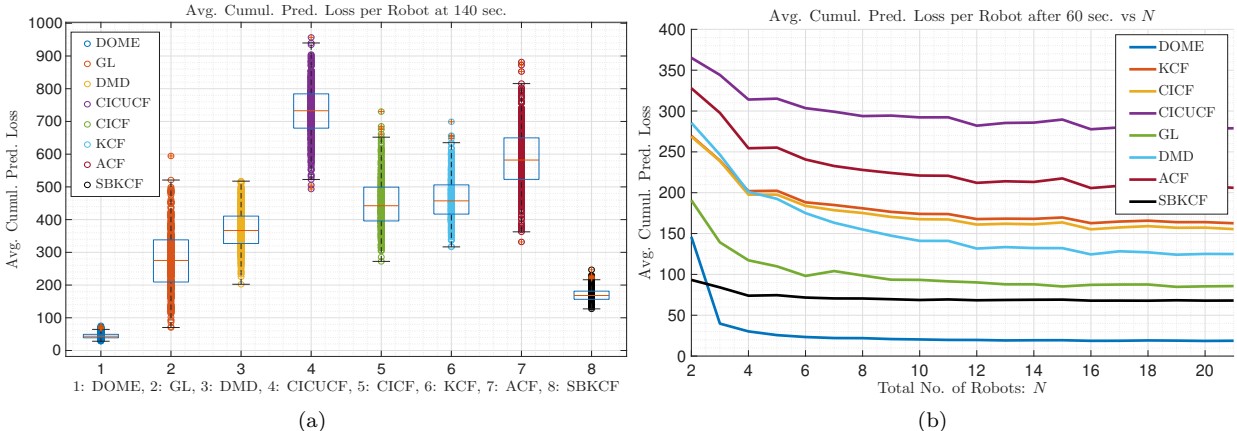

Figure 3: (a) A box plot showing the average cumulative 10-step look-ahead prediction loss per robot at 140 seconds; 500 simulation runs (points) for each method – 1: DOME, 2: GL, 3: DMD, 4: CICUCF, 5: CICF, 6: KCF, 7: ACF, 8: SBKCF. (b) Average cumulative 10-step look-ahead prediction loss per robot at 60 sec. as a function of total no. of robots $N$, averaged over 100 simulation runs for each method.

as $\mathbf{C}_{t,i}^{\tau} = (1 + \tau \cdot prj\_rate) \cdot \mathbf{C}_{t,i}^{1}$. This setup reflects scenarios where noise and bias are not necessarily correlated.

*Remark* 9. The DOME algorithm makes no structural assumptions about bias dynamics. As stated in Section 3 (Abstract Model for Prediction Algorithm), the bias is modeled as arbitrary and unknown, potentially non-Gaussian, and without a known distribution; this arbitrariness also allows for adversarial bias. DOME employs a universal online learning update that does not depend on internal physics models (e.g., random-walk or constant-bias models typically required by Kalman filter variants). The "structured" switching-bias model used in this section (Fig. 2a) serves only to generate synthetic data for stress-testing under abrupt failures, and DOME was not given access to this structure.

Under this setup, DOME fusion employs learning rates $\eta_w = 10$ and $\eta_\alpha = 4$, with a periodic reset interval $T_o = 25$, as determined through a simulation-based parametric study (see Appendix A.3.3 for more details).

Under the described communication and prediction uncertainties, DOME is evaluated against four covariance-based methods – Kalman Consensus Fusion (KCF), Covariance Intersection Consensus Fusion (CICF), Covariance Intersection + Covariance Union Consensus Fusion (CICUCF), and Separate Bias Kalman Consensus Fusion (SBKCF), and two online learning-based methods – Greedy-Local (GL) and Distributed Mirror Descent (DMD). SBKCF is inspired by the Separate Bias Kalman Filter (SBKF) (Friedland, 1969; Ignagni, 1990). DMD is applied by modifying the mirror descent-based distributed online optimization algorithm proposed by Shahrampour & Jadbabaie (2017a;b) to fit the problem setting considered in this paper. An Averaging Consensus Fusion (ACF) baseline is also included. A detailed description of these methods is provided in the Appendix A.3.2.

These algorithms are evaluated based on average cumulative $\tau$-step look-ahead prediction loss for $t \geq \tau$ : $\frac{1}{N} \sum_{i=1}^{N} \sum_{s=\tau}^{t} l(\hat{\mathbf{x}}_{(s|s-\tau),B}^{i}, \mathbf{x}_{s,B})$, with $\tau = 10$, $N = 6$.

Figure 3a presents a box plot of the average cumulative 10-step look-ahead prediction loss per robot at 140 seconds, aggregated over 500 simulation runs for each method. In each box, the central line denotes the median, while the bottom and top edges represent the $25^{th}$ and $75^{th}$ percentiles, respectively. Notably, DOME achieves a substantially lower prediction loss compared to all other fusion algorithms. In terms of median performance, DOME reduces the average cumulative prediction loss by approximately 74% relative to the next best method, SBKCF. In addition to superior accuracy, DOME also demonstrates markedly lower variance, indicating greater consistency across simulation runs.

DOME fusion is further evaluated for scalability by measuring the average cumulative 10-step look-ahead prediction loss per robot at the end of a 60-second horizon, averaged over 100 simulation runs, as a function

of the total number of robots $N$ (see Fig. 3b). Starting from $N = 2$, robots are incrementally added to the underlying linear graph structure without introducing random link drops. At times 0.0, 23.33, 46.67, 70, and 116.67 seconds, robots are randomly assigned either $c_{t,i} = 0.01$ or $c_{t,i} = 4$ (see Fig. 2a), such that at least $\lfloor \frac{N}{2} \rfloor$ and at most $\lfloor \frac{N}{2} \rfloor + 1$ out of the total $N$ robots use inaccurate prediction algorithms ($c_{t,i} = 4$), where $\lfloor \cdot \rfloor$ denotes the floor function.

The noise covariance behavior remains consistent with the previous simulation setup, representing scenarios where noise and bias are not necessarily correlated. As shown in Fig. 3b, DOME outperforms all other fusion algorithms in this scalability test. Its average cumulative 10-step look-ahead prediction loss per robot remains substantially lower – approximately 72.4% less than the next best algorithm, SBKCF – as $N$ increases. Moreover, beyond $N = 10$, DOME's performance remains nearly constant, indicating that the algorithm maintains high reliability and prediction accuracy even as the swarm size grows. This suggests that DOME enables scalable swarm behavior without compromising predictive performance.

From Figures 3a and 3b, it is evident that DOME outperforms the other fusion methods evaluated in the simulation studies. Notably, DOME, GL, and DMD do not require knowledge of prediction covariances and operate without assumptions of consistency or cross-correlation. Consequently, they outperform covariance-based methods – KCF, CICF, and CICUCF – which are sensitive to large prediction biases, particularly when such biases are not reflected in inflated covariance estimates.

SBKCF, however, achieves the strongest baseline performance due to its drift estimation mechanism inspired by the Separate Bias Kalman Filter (SBKF). By adopting a random-walk bias model, SBKCF explicitly predicts and compensates for future drift, thereby correcting biased onboard predictions. In the scalability study (Fig. 3b), this modeling advantage allows SBKCF to outperform DOME for $N = 2$, where explicit drift estimation is beneficial. However, for $N \geq 3$, the same modeling assumption becomes restrictive: reliance on a predefined internal drift model limits robustness under non-stationary environments with unknown or arbitrary drift dynamics. In contrast, DOME employs a model-agnostic online learning update that does not depend on internal physics assumptions (e.g., random-walk or constant-bias models), enabling greater adaptability in heterogeneous and dynamically evolving settings.

Although both ACF and DMD assign equal weights to information from neighboring robots, DMD achieves better performance due to its online error correction mechanism. In contrast, DOME surpasses both DMD and GL by employing an adaptive weighting scheme based on an online learning process. Unlike DOME, neither ACF nor DMD includes a mechanism for adaptively filtering out biased predictions from other robots in the network. Consequently, biased information can propagate unchecked through the communication network, degrading the performance of even those robots with accurate prediction algorithms. GL outperforms DMD by greedily selecting the most accurate local prediction; however, it does not propagate this selection across the network.

The MATLAB simulation video is submitted as a supplementary file. Furthermore, DOME is implemented in a Gazebo simulation environment using ROS (a simulation video is submitted as a supplementary file).

**Ablation Study**: A simulation-based ablation study of DOME (Appendix A.3.5, Fig. 6) further clarifies the relative contributions of its components. The results indicate that adaptive weighting is more influential in the local fusion layer than in the social layer, whereas the social fusion layer itself has a greater overall effect on performance than the local fusion layer. The periodic reset mechanism, meanwhile, emerges as an essential component of the framework.

**Robustness Study**: Further, DOME is evaluated under varying communication link drop probabilities (Appendix A.3.6, Fig. 7) and prediction bias/drift rates (Appendix A.3.7, Fig. 8), demonstrating that it maintains acceptable performance – remaining within the 100-loss median range – up to $p_{ld} = 0.5$ and $c_{t,i} = 10$. DOME is further evaluated under varying observation noise magnitudes $\|\epsilon_t\|_2$ (Appendix A.5, Fig. 9). The results indicate that it maintains acceptable performance – remaining within the 100-loss median range – up to $\|\epsilon_t\|_2 = 10$ meters. A detailed theoretical and empirical discussion regarding the impact of noisy observations is provided in the Appendix A.5.

**Sensitivity to Mismatch in** $T_o$: The convergence analysis in Theorem 1 applies within each stationary epoch (i.e., between resets), implying that optimal performance is achieved when $T_o$ is commensurate with

the environment's switching interval. If $T_o$ is too small, the algorithm may remain in a transient high-variance regime; if too large, adaptivity lag can occur after a regime shift. However, empirical results indicate a broad performance basin rather than sharp sensitivity. In the event-driven stochastic bias model (Section 6, Fig. 2a), DOME maintains strong performance across a wide range of fixed $T_o$ values (Appendix A.3.3, Table 1), with only gradual degradation away from the optimal setting. A detailed theoretical and empirical discussion is provided in the Appendix A.6.

## 7    Conclusion

This paper presents a Distributed Online learning-based Multi-Estimate (DOME) fusion algorithm for cooperative predictive target tracking in a robotic swarm subject to dynamic prediction bias and communication uncertainty. Designed for heterogeneous teams with diverse sensors and prediction models, DOME performs a two-layer weighted fusion of local and socially shared predictions, combined with an implicit, adaptive, consensus-like update. Fusion weights are learned online using prediction loss feedback, enabling each robot to improve its own predictions while supporting its neighbors. Theoretical analysis shows convergence of the expected learning weights under reasonable assumptions. DOME is computationally lightweight and analytically tractable, making it practical for swarm deployment. Simulations under conditions of large prediction drift and random link failures show that DOME outperforms both covariance-based and online learning-based decentralized methods, achieving a 74% reduction in prediction loss compared to the next-best method, Separate Bias Kalman Consensus Fusion (SBKCF). It also scales well, with a 72.4% lower prediction loss than SBKCF as the swarm size increases. Finally, DOME is validated in a ROS-Gazebo simulation environment (check the simulation video in the supplementary zip file). This study assumes full observability of the target; future work will address the partially observable case.

### Broader Impact Statement

The proposed DOME fusion algorithm enhances collaborative prediction accuracy in decentralized multi-agent systems operating under communication uncertainty and prediction biases, with particular relevance to multi-drone systems, where reliable distributed prediction and fusion are critical for applications such as surveillance, environmental monitoring, disaster response, and autonomous navigation. For responsible deployment, strict adherence to ethical guidelines is essential when operating autonomous monitoring systems in populated areas.

While developed for cooperative predictive target tracking using a robotic swarm, the DOME framework generalizes across domains requiring decentralized, privacy-preserving, and adaptive prediction fusion. Potential applications span finance, where distributed trading models collaboratively forecast market trends; weather and climate forecasting, where regional prediction centers integrate heterogeneous models; urban sensing and environmental monitoring, where sensor networks fuse multi-modal data; and healthcare, where institutions jointly improve predictive diagnostics without sharing sensitive data. In smart grids and supply chains, DOME can enable robust coordination among nodes by cooperatively predicting energy availability and demand under uncertainty.

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

# A Appendix

## A.1

## Nomenclature

| | |
|---|---|
| $\lVert \cdot \rVert$ | 2-norm |
| $^{n}C_{k}$ | $n$ choose $k$ |
| $(\cdot)'$ | transpose operation |
| $\lvert \cdot \rvert$ | based on usage can either be the cardinality of a set or element-wise absolute value operation for a real-valued vector |
| $\mathbb{E}[\cdot]$ | expectation operator |
| $p_{ld}$ | communication link-drop probability |
| $T$ | time horizon – discrete-time |
| $T_o$ | periodic reset in DOME occurs after every $T_o$ discrete-time steps |
| $\mathbf{1}(\cdot)$ | the indicator function |
| $\lfloor \cdot \rfloor$ | the floor function |
| $\log(\cdot)$ | natural log |
| $\Pi(\cdot)$ | $\Pi_{m=1}^{n_o}(a_m) = a_1 a_2 a_3 \cdots a_{n_o}$ |
| $N$ | total number of robots in the swarm |
| $G(t)$ | undirected random communication connectivity graph at time $t$ |
| $\Omega_{t,i}$ | $i^{th}$ robot's neighbor set as per $G(t)$ at time $t$ |
| $N_i(t)$ | $i^{th}$ robot's total number of neighbors at time $t$, $N_i(t) = \lvert \Omega_{t,i} \rvert$ |
| $\bar{G}$ | base graph for the undirected random graph $G(t)$ |
| $\bar{\Omega}_i$ | neighbor set of the $i^{th}$ robot as per the base-graph $\bar{G}$ |
| $n_i$ | maximum possible neighbors of the $i^{th}$ robot as per the base-graph $\bar{G}$, $n_i = \lvert \bar{\Omega}_i \rvert$ |
| $A_i$ | $i^{th}$ robot's prediction algorithm |
| $\mathbf{x}_{t,i}$ | $i^{th}$ robot's position vector (in $m$) |
| $\mathbf{x}_{t,B}$ | target's (Bogey's) position vector (in $m$) |
| $\zeta_{t,i}^{\tau}$ | drift in algorithm $A_i$'s $\tau$-step look-ahead prediction |
| $\hat{\mathbf{x}}_{(t+\tau\mid t),B}^{A_i}$ | $\tau$-step look-ahead prediction of target's position given by the $i^{th}$ robot's algorithm $A_i$ |
| $\hat{\mathbf{x}}_{(t+\tau\mid t),B}^{L_i}$ | $i^{th}$ robot's Local $\tau$-step look-ahead prediction of target's position given by the $i^{th}$ robot's algorithm $A_i$ |
| $\hat{\mathbf{x}}_{(t+\tau\mid t),B}^{i}$ | $i^{th}$ robot's Social $\tau$-step look-ahead prediction of target's position |
| $l(\cdot,\cdot)$ | bounded loss function with two arguments; $l(\cdot,\cdot) \in [0,1]$ |
| $l_{t,i}^{A}$ | algorithm $A_i$'s prediction loss: $l(\hat{\mathbf{x}}_{(t\mid t-1),B}^{A_i}, x_{t,B})$ |
| $L_{t,i}^{A}$ | algorithm $A_i$'s cumulative prediction loss: $\sum_{s=1}^{t} l_{s,i}$ |
| $l_{t,i}^{L}$ | $i^{th}$ robot's Local prediction loss: $l(\hat{\mathbf{x}}_{(t\mid t-1),B}^{L_i}, x_{t,B})$ |
| $L_{t,i}^{L}$ | $i^{th}$ robot's Local prediction's cumulative loss: $\sum_{s=1}^{t} l_{s,i}^{L}$ |
| $l_{t,i}^{S}$ | $i^{th}$ robot's Social prediction loss: $l(\hat{\mathbf{x}}_{(t\mid t-1),B}^{i}, x_{t,B})$ |
| $L_{t,i}^{S}$ | $i^{th}$ robot's Social prediction's cumulative loss: $\sum_{s=1}^{t} l_{s,i}^{S}$ |
| $l_{t,i}^{S-}$ | $i^{th}$ robot's previous-time Social prediction loss w.r.t. current target position: $l(\hat{\mathbf{x}}_{(t-1\mid t-2),B}^{i}, x_{t,B})$ |
| $L_{t,i}^{S-}$ | $i^{th}$ robot's previous-time Social prediction's cumulative loss w.r.t. current target position: $\sum_{s=1}^{t} l_{s,i}^{S-}$ |
| $\mathcal{H}_{t,i}$ | history relevant to the $i^{th}$ robot just before the communication phase begins, at time $t$ |

$\Omega_t^{k,l}$    one of the many possible neighbor sets containing $k$ neighboring robots that the $i^{th}$ robot is communicating with at time $t$, where $k \in \{0, 1, 2, \cdots, n_i\}$, and $l \in \{1, 2, \cdots, {}^{n_i}C_k\}$

$\Lambda_{t,i}$    $\Omega_{t,i} \cup \{i\}$

$\bar{\Lambda}_i$    $\bar{\Omega}_i \cup \{i\}$

$\Lambda_{t,i}^{k,l}$    $\Omega_{t,i}^{k,l} \cup \{i\}$, where $k \in \{0, 1, 2, \cdots, n_i\}$, and $l \in \{1, 2, \cdots, {}^{n_i}C_k\}$

$\eta_\alpha, \eta_w$    DOME algorithm's learning rate parameters

$n_{max}$    max. limit on the no. of neighbors any robot can have in the base graph $\bar{G}$, such that $n_i \leq n_{max}$

$\alpha_i(t)$    $i^{th}$ robot's Local trust weight at time $t$; $\alpha_i(t) \in [0, 1]$

$\bar{v}_{t,i}$    $i^{th}$ robot's body-axis velocity vector (in $m/s$)

$\bar{w}_{t,i}$    $i^{th}$ robot's yaw rate (in $rad/s$)

$\Delta T$    sampling period (in $s$)

$\epsilon, \beta$    positive constants governing the growth rate of cumulative loss differences (Assumption 2); $\epsilon \in (0, 1], \beta \in (0, 1]$

$\hat{\alpha}_i(t), \hat{\alpha}'_i(t)$    unnormalized weight factors used for updating the Local trust weight $\alpha_i(t)$

$\hat{w}_{ii}(t)$    $i^{th}$ robot's unnormalized self-trust weight

$\mu_{t,i}^\tau$    bias component of the prediction drift $\zeta_{t,i}^\tau$

$\nu_{t,i}^\tau$    zero-mean Gaussian noise component of the prediction drift $\zeta_{t,i}^\tau$

$\phi_{t,i}$    $i^{th}$ robot's heading angle (in $rad$)

$c_{t,i}$    scalar parameter indicating the accuracy of algorithm $A_i$ in the simulation experiments

$C_{t,i}^\tau$    covariance matrix of the prediction noise $\nu_{t,i}^\tau$ (in $m^2$)

$j_{*,i}^{k,l}(t)$    index of the best-performing robot (minimum cumulative loss) within the subset $\Lambda_i^{k,l}$ at time $t$

$j_{\infty,i}^{k,l}$    unique limit of the index $j_{*,i}^{k,l}(t)$ as $t \to \infty$ (Assumption 1)

$J_i$    set of indices of all robots that are asymptotically best for robot $i$ across all possible neighbor subsets

$M_{j,i}$    set of neighbor configuration pairs $(k, l)$ for which robot $j$ is the asymptotically best source for robot $i$

$w_{ij}(t)$    Social trust weight assigned by robot $i$ to robot $j$ at time $t$; $w_{ij}(t) \in [0, 1]$

$w_{ij}^{k,l}(t)$    intermediate weight term used in the expectation analysis of $w_{ij}(t)$ conditioned on the neighbor set $\Lambda_i^{k,l}$

## A.2 Decentralized Normalization Scheme

The weights are normalized using a decentralized normalization scheme summarized as Algorithm 2. A similar procedure is used for the normalization of the weights $\hat{\alpha}_i(t)$ and $\hat{\alpha}'_i(t)$ as well.

## A.3 MATLAB Simulation Details

In the simulations, the target $(B)$ follows a unicycle model with $i = B$, $\bar{v}_{t,B}^y = 0$, where $\bar{\mathbf{v}}_{t,B} = [\bar{v}_{t,B}^x \ \bar{v}_{t,B}^y]'$ and $\bar{v}_{t,B}^x \in \mathbb{R}$, $\bar{v}_{t,B}^y \in \mathbb{R}$. The target changes its speed after every 5 *sec.* intervals, such that $\bar{v}_{t,B}^x$ is sampled as $\bar{v}_{t,B}^x \sim (8 + Unif.(0, 4))$ $m/s$ after every 5 seconds. Further, the target also changes its yaw rate $(\bar{w}_{t,B})$ after every 5 *sec.* intervals, such that $\bar{w}_{t,B}$ ($rad./s$) is sampled from either of the three expressions with equal probability: $-\frac{6\pi}{T \cdot \Delta T} - Unif.\left(0, \frac{6\pi}{T \cdot \Delta T}\right)$, $0$, and $\frac{6\pi}{T \cdot \Delta T} + Unif.\left(0, \frac{6\pi}{T \cdot \Delta T}\right)$. The initial position $(\mathbf{x}_{0,B})$ and yaw angle $(\psi_{0,B})$ of the target are randomly sampled as $\mathbf{x}_{0,B} \sim [Unif.(0, 20) \ Unif.(0, 20)]'$ $m$ and $\psi_{0,B} \sim Unif.(-\pi, \pi)$ $rad.$, respectively, at the start of each simulation run.

For the robots, the velocity control input is bounded as $|\bar{\mathbf{v}}_{t,i}| \leq [10, 10]'$ $m/s$, and the yaw-rate control input is bounded as $|\bar{w}_{t,i}| \leq 0.524$ $rad./s$, where $|\cdot|$ is the element-wise absolute value operation. The initial position $\mathbf{x}_{t,i}$ and yaw angle $\psi_{t,i}$ of the $i^{th}$ robot is set to be $\mathbf{x}_{t,i} = [10 \cdot i, 0]'$ $m$ and $\psi_{t,i} = \pi/2$ $rad.$, respectively. The control law parameters are set to be $d_S = 10$ $m$, $k_1 = 4$, $k_2 = 10$, and $k_3 = 10$. The loss function is defined to be $l(\mathbf{x}, \mathbf{y}) = \min(||\mathbf{x} - \mathbf{y}||/50, 1)$, where $\mathbf{x}, \mathbf{y} \in \mathbb{R}^2$.

---

**Algorithm 2** Decentralized normalization scheme for the $i^{th}$ robot at time $t$, $\forall i \in [N]$

---

    **Choose:** machine's least precision $\delta > 0$
    **Initialize:** $nrmcnt_{ii} = 0$
1: **if** $(\hat{w}_{ii}(t) \leq \delta)$ **then**
2:     $\hat{w}_{ii}(t) \leftarrow \hat{w}_{ii}(t)/\delta$
3:     $nrmcnt_{ii} \leftarrow nrmcnt_{ii} + 1$
4: **end if**
5: send $\{i, \hat{w}_{ii}(t), nrmcnt_{ii}\}$ to and receive $\{j, \hat{w}_{jj}(t), nrmcnt_{jj}\}$ from neighbors $j \in \Omega_{t,i}$
6: $\hat{w}_{ij}(t) = \begin{cases} \hat{w}_{jj}(t) & : & \forall j \in \Lambda_{t,i} \\ 0 & : & otherwise \end{cases}$
7: **for** $(j \in \Lambda_{t,i})$ **do**
8:     **if** $(nrmcnt_{jj} > \min_{j' \in \Lambda_{t,i}} nrmcnt_{j'j'})$ **then**
9:         $\hat{w}_{ij}(t) \leftarrow 0$
10:     **end if**
11: **end for**

---

### A.3.1 Control Law

**Translational Control Law:** for the $i^{th}$ robot, the translational control law consists of two terms as given below

$$\bar{\mathbf{v}}_{t,i} = \bar{\mathbf{v}}_{t,i}^R + \Delta\bar{\mathbf{v}}_{t,i} \tag{29}$$

where $\bar{\mathbf{v}}_{t,i}^R$ is the $i^{th}$ robot's reference command signal responsible for chasing the target, and $\Delta\bar{\mathbf{v}}_{t,i}$ is the $i^{th}$ robot's correction control signal responsible for avoiding collisions with other robots.

Denote $\mathbf{R}_{t,i} \in \mathbb{R}^{2 \times 2}$ as the $i^{th}$ robot's body-global rotation matrix at time $t$, defined as $\mathbf{R}_{t,i} = \begin{bmatrix} \cos\phi_{t,i} & -\sin\phi_{t,i} \\ \sin\phi_{t,i} & \cos\phi_{t,i} \end{bmatrix}$.

The $i^{th}$ robot's reference command signal $\bar{\mathbf{v}}_{t,i}^R$ is given as

$$\bar{\mathbf{v}}_{t,i}^R = k_1 \mathbf{R}_{t,i}' \frac{\Delta\hat{\mathbf{x}}_{(t+\tau|t),B}^i}{||\Delta\hat{\mathbf{x}}_{(t+\tau|t),B}^i||} (||\Delta\mathbf{x}_{t,B}^i|| - d_S) \tag{30}$$

where $(\cdot)'$ represents the transpose operation, $||\cdot||$ is the 2-norm, $k_1 > 0$ is a control parameter. $\Delta\mathbf{x}_{t,B}^i := \mathbf{x}_{t,B} - \mathbf{x}_{t,i}$, where $\mathbf{x}_{t,B}$ is the target's position vector at time $t$, and $\mathbf{x}_{t,i}$ is the $i^{th}$ robot's position vector at time $t$. $d_S > 0$ $(m)$ indicates the distance each robot should maintain from the target while chasing it. Here, $\Delta\hat{\mathbf{x}}_{(t+\tau|t),B}^i$ is defined as

$$\Delta\hat{\mathbf{x}}_{(t+\tau|t),B}^i := \hat{\mathbf{x}}_{(t+\tau|t),B}^i - \mathbf{x}_{t,i} \tag{31}$$

where $\hat{\mathbf{x}}_{(t+\tau|t),B}^i$ is $i^{th}$ robot's $\tau$-step look-ahead prediction of target's position at time $t$, and $\mathbf{x}_{t,i}$ is $i^{th}$ robot's position at time $t$.

Further, we assume that each robot is equipped with a collision avoidance system, which ensures that while chasing the target, robots do not collide. Considering eq.(29), this behavior can be modeled by the correction control signal $\Delta\bar{\mathbf{v}}_{t,i}$ for the $i^{th}$ robot by using an *inter-robot collision avoidance* control law given as follows:

$$\Delta\bar{\mathbf{v}}_{t,i} = -k_2 \mathbf{R}_{t,i}' \frac{\mathbf{x}_{t,p_t^i} - \mathbf{x}_{t,i}}{||\mathbf{x}_{t,p_t^i} - \mathbf{x}_{t,i}||^2} \tag{32}$$

where $(\cdot)'$ represents the transpose operation, $||\cdot||$ is the 2-norm or the Euclidean norm, $k_2 > 0$ is a control parameter, $p_t^i \in [N] \setminus \{i\}$ is the index of the robot spatially nearest to $i^{th}$ robot at time $t$, formally defined as $p_t^i := \arg\min_{j \in [N] \setminus \{i\}} ||\mathbf{x}_{t,j} - \mathbf{x}_{t,i}||$. Thus, $\mathbf{x}_{t,p_t^i}$ is the position vector of the robot spatially nearest to the $i^{th}$ robot at time $t$.

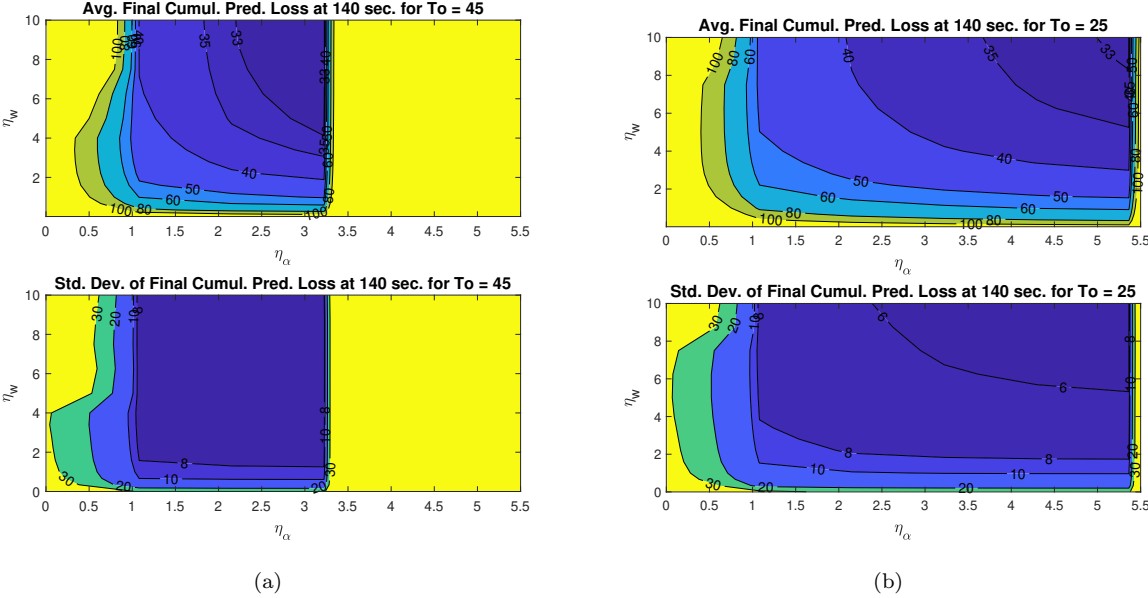

Figure 4: The mean and standard deviation of the average social cumulative prediction loss of all the robots at $t = 140$ *sec.* (averaged over 50 sim. runs) versus the learning rates for: (a) $T_o = 45$ and (b) $T_o = 25$.

**Heading Control Law:** consider a heading angle requirement such that the robots are required to yaw in a way that their heading direction should point towards their 1-step look-ahead estimate of the target's position $\hat{\mathbf{x}}^i_{(t+1|t),B}$. The angle between $\Delta\hat{\mathbf{x}}^i_{(t+1|t),B}$ (from eq.31) and the $i^{th}$ robot's heading direction $\mathbf{h}_{t,i} = \begin{bmatrix}\cos\phi_{t,i} & \sin\phi_{t,i}\end{bmatrix}'$, with respect to the $\Delta\hat{\mathbf{x}}^i_{t+1,B}$ direction, can be obtained as $\Delta\phi^i_{t,err} = atan2\left(\mathbf{h}_{t,i} \times \Delta\hat{\mathbf{x}}^i_{t+1,B}, \mathbf{h}_{t,i} \cdot \Delta\hat{\mathbf{x}}^i_{t+1,B}\right)$, where the first argument involves a cross-product and the second argument involves dot-product. As per the heading angle requirement, $i^{th}$ robot's yaw control law can be given as

$$\bar{w}_{t,i} = k_3\Delta\phi^i_{t,err} \tag{33}$$

where $k_3 > 0$ is a control parameter.

### A.3.2   Baselines

For the above-described simulation setup with an adverse setting, DOME is compared against four covariance-based (Kalman Consensus Fusion, Covariance Intersection Consensus Fusion, Covariance Intersection + Covariance Union Consensus Fusion, Separate Bias Kalman Consensus Fusion) and two online learning-based (Greedy-Local, Distributed Mirror Descent) decentralized prediction fusion methods, along with the Averaging Consensus Fusion baseline, which are briefly described as follows:

**Averaging Consensus Fusion (ACF)**: instead of updating the weights $\hat{\alpha}_i(t)$, $\hat{\alpha}'_i(t)$, and $\hat{w}_{ii}(t)$ as per the DOME fusion weight update mechanism, keep them fixed as $\hat{\alpha}_i(t) = 1$, $\hat{\alpha}'_i(t) = 1$, and $\hat{w}_{ii}(t) = 1$ and calculate the predictions.

**Kalman Consensus Fusion (KCF)** (Maybeck, 1982; Uhlmann, 2003): assumes that the predictions being fused are uncorrelated and their associated zero-mean Gaussian noises' covariance ($\mathbf{C}^\tau_{t,i}$) are known.

Consider the covariance of $\hat{\mathbf{x}}^{L_i}_{(t+\tau|t),B}$ and $\hat{\mathbf{x}}^i_{(t-1+\tau|t-1),B}$ as $\mathbf{C}^{L_i}_{(t+\tau|t),B}$ and $\mathbf{C}^i_{(t-1+\tau|t-1),B}$, respectively. Consider $\mathbf{C}^\tau_{t,i}$ as the covariance for $\hat{\mathbf{x}}^{A_i}_{(t+\tau|t),B}$.

*Local prediction phase:*

$$(\mathbf{C}^{L_i}_{(t+\tau|t),B})^{-1} = \frac{1}{2}\left((\mathbf{C}^i_{(t-1+\tau|t-1),B})^{-1} + (\mathbf{C}^\tau_{t,i})^{-1}\right) \tag{34}$$

| $T_o$ | $\eta_w$ | $\eta_\alpha$ | $\min_{\eta_\alpha,\eta_w}\{\frac{1}{6}\sum_{i=1}^6 L_{1400,i}^S\}$ |
|---|---|---|---|
| 5 | 10 | 15 | 76.62 |
| 10 | 10 | 13.9 | 49.49 |
| 15 | 10 | 9.6 | 40.30 |
| 20 | 10 | 6.4 | 36.33 |
| 25 | 10 | 5.4 | 32.56 |
| 30 | 10 | 4.3 | 32.75 |
| 35 | 8.8 | 3.2 | 34.54 |
| 40 | 7.5 | 3.2 | 33.15 |
| 45 | 8.8 | 3.2 | 30.55 |
| 50 | 10 | 2.2 | 34.05 |
| 55 | 10 | 2.2 | 34.59 |
| 60 | 6.3 | 2.2 | 34.55 |
| 65 | 7.5 | 2.2 | 34.48 |
| 70 | 8.8 | 1.1 | 42.71 |
| 75 | 6.3 | 1.1 | 37.56 |
| 80 | 6.3 | 1.1 | 41.28 |
| 85 | 6.3 | 1.1 | 42.93 |
| 90 | 6.3 | 1.1 | 39.62 |
| 95 | 6.3 | 1.1 | 40.77 |
| 100 | 6.3 | 1.1 | 42.30 |

Table 1: Optimal learning rates for different $T_o$ values from a simulation-based parametric study of DOME.

$$\hat{\mathbf{x}}_{(t+\tau|t),B}^{L_i} = \left((\mathbf{C}_{(t-1+\tau|t-1),B}^i)^{-1} + (\mathbf{C}_{t,i}^\tau)^{-1}\right)^{-1}\left((\mathbf{C}_{t,i}^\tau)^{-1}\hat{\mathbf{x}}_{(t+\tau|t),B}^{A_i} + (\mathbf{C}_{(t-1+\tau|t-1),B}^i)^{-1}\hat{\mathbf{x}}_{(t-1+\tau|t-1),B}^i\right) \tag{35}$$

*Communication phase:* The $i^{th}$ robot sends the information $\{t, i, \hat{\mathbf{x}}_{(t+\tau|t),B}^{L_i}, \mathbf{C}_{(t+\tau|t),B}^{L_i}\}$ and receives the information $\{t, j, \hat{\mathbf{x}}_{(t+\tau|t),B}^{L_j}, \mathbf{C}_{(t+\tau|t),B}^{L_j}\}$ from its communicating neighbors $j \in \Omega_{t,i}$.

*Social prediction phase:* With $n_{t,i} = |\Omega_{t,i}|$, we have

$$(\mathbf{C}_{(t+\tau|t),B}^i)^{-1} = \frac{1}{n_{t,i}+1}\sum_{\forall j \in \Lambda_{t,i}}(\mathbf{C}_{(t+\tau|t),B}^{L_j})^{-1} \tag{36}$$

$$\hat{\mathbf{x}}_{(t+\tau|t),B}^i = \left(\sum_{\forall j \in \Lambda_{t,i}}(\mathbf{C}_{(t+\tau|t),B}^{L_j})^{-1}\right)^{-1}\sum_{\forall j \in \Lambda_{t,i}}(\mathbf{C}_{(t+\tau|t),B}^{L_j})^{-1}\hat{\mathbf{x}}_{(t+\tau|t),B}^{L_j} \tag{37}$$

**Covariance Intersection Consensus Fusion (CICF)** (Matzka & Altendorfer, 2009; Julier & Uhlmann, 2017): consider the equations (34) – (37); each robot employs the Covariance Intersection (CI) method for the fusion of the predictions instead of Kalman Fusion. CI assumes that the predictions being fused are consistent and their associated zero-mean Gaussian noises' covariance ($\mathbf{C}_{t,i}^\tau$) is known, but their cross-correlation is unknown.

**Covariance Intersection + Covariance Union Consensus Fusion (CICUCF)** (Matzka & Altendorfer, 2009; Reece & Roberts, 2010): consider the equations (34) – (37); first, each robot employs the Covariance Intersection (CI) method for the fusion in the local prediction phase instead of Kalman fusion. After this, the output of CI is fused with the CI-predictions from the neighbors in the social prediction phase using Covariance Union (CU) instead of Kalman fusion. In CU, while trying to keep the fused predictions consistent, the resultant fused covariance is increased. Therefore, CI is applied in the next iteration to reduce the predictions' covariance. CU assumes that the predictions being fused can be inconsistent and their cross-correlation is unknown, but their associated zero-mean Gaussian noises' covariance ($\mathbf{C}_{t,i}^\tau$) is known.

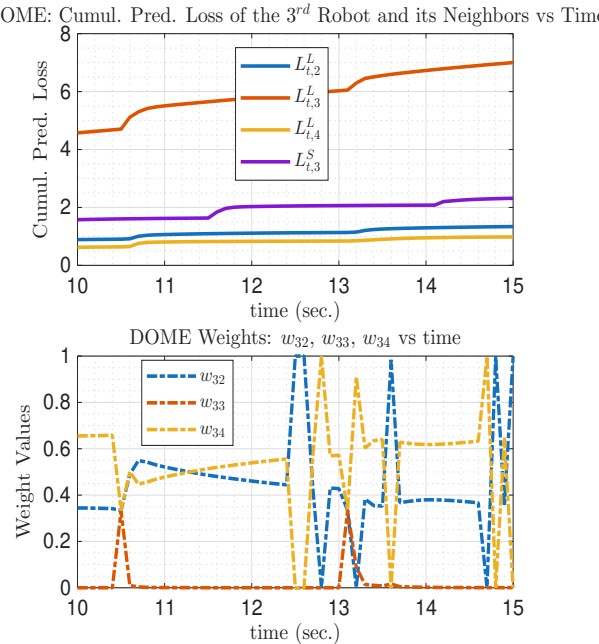

Figure 5: The first plot shows the cumulative prediction losses for selected robots for a duration of 5 seconds. The second plot illustrates how the weights change in response to the prediction losses shown in the first plot; note that random link drops occur, which means that the nominal neighbors may not be neighbors at all times.

**Greedy-Local (GL)**: each robot uses the one-step look-ahead prediction $\hat{\mathbf{x}}^i_{(t+1|t),B} = \hat{\mathbf{x}}^{A_{j_*}}_{(t+1|t),B}$ and the $\tau$-step look-ahead prediction $\hat{\mathbf{x}}^i_{(t+\tau|t),B} = \hat{\mathbf{x}}^{A_{j_*}}_{(t+\tau|t),B}$, which incurs the least cumulative loss among all the predictions that are shared by its neighbors and its own prediction algorithm, i.e., $j_* = \arg\min_{j \in \Lambda_{t,i}} L^A_{t,j}$, $\forall i \in [N]$. Here, $l^A_{t,i} = l(\hat{\mathbf{x}}^{A_i}_{(t|t-1),B}, \mathbf{x}_{t,B})$ and $L^A_{t,i} = \sum_{s=1}^t l^A_{s,i}$.

**Distributed Mirror Descent (DMD)**: Mirror Descent is applied to our problem setting by modifying the Mirror Descent-based distributed online optimization algorithm proposed by Shahrampour & Jadbabaie, 2017a;b, leading to the following:

$$\hat{\mathbf{x}}^i_{(t|t),B} = \sum_{\forall j \in \Lambda_{t,i}} W_{ij}(t)[\hat{\mathbf{x}}^j_{(t|t-1),B} + \eta_m(\mathbf{x}_{t,B} - \hat{\mathbf{x}}^{A_j}_{(t|t-1),B})] \tag{38}$$

$$\hat{\mathbf{x}}^i_{(t+1|t),B} = \hat{\mathbf{x}}^i_{(t|t),B} + \sum_{\forall j \in \Lambda_{t,i}} W_{ij}(t)(\hat{\mathbf{x}}^{A_j}_{(t+1|t),B} - \hat{\mathbf{x}}^{A_j}_{(t|t-1),B}) \tag{39}$$

$$\hat{\mathbf{x}}^i_{(t+\tau|t),B} = \hat{\mathbf{x}}^i_{(t|t),B} + \sum_{\forall j \in \Lambda_{t,i}} W_{ij}(t)(\hat{\mathbf{x}}^{A_j}_{(t+\tau|t),B} - \hat{\mathbf{x}}^{A_j}_{(t|t-1),B}) \tag{40}$$

Here, $\eta_m > 0$ is a learning parameter, and $W_{ij}(t) \in \mathbb{R}_{\geq 0}$ are the weights for the consensus-based fusion. The weights $W_{ij}(t)$ are considered to be equal for all the neighbors at any time $t$, i.e., $W_{ij}(t) = (n_{t,i} + 1)^{-1}$, where $n_{t,i} = |\Omega_{t,i}|$, $\forall j \in \Lambda_{t,i}$.

**Separate Bias Kalman Consensus Fusion (SBKCF)**: applies the principle of bias separation (Friedland, 1969; Ignagni, 1990) to Kalman Consensus Fusion (KCF), and assumes that the predictions being fused are uncorrelated and their associated zero-mean Gaussian noises' covariance ($\mathbf{C}^{\tau}_{t,i}$) are known. The bias or drift $\zeta^1_{t,i}$ is modeled as following a Random Walk process to enable the prediction of future bias estimates.

Calculate the residual at time $t$: $\mathbf{y}_{t,i} = \hat{\mathbf{x}}^{A_i}_{(t|t-1),B} - \mathbf{x}_{t,B}$. From equation (3), note that $\zeta^1_{t-1,i} = \mathbf{y}_{t,i}$. The drift $\zeta^1_{t,i}$ is assumed to follow a Random Walk process as follows:

$$\zeta^1_{t,i} = \zeta^1_{t-1,i} + \omega^{\text{drift}}_t \tag{41}$$

where $\omega_t^{\text{drift}} \sim \mathcal{N}(0, \mathbf{Q}_t^{\text{drift}})$, i.e., $\omega_t^{\text{drift}} \in \mathbb{R}^2$ (in meters) is a zero-mean Gaussian noise with covariance $\mathbf{Q}_t^{\text{drift}} \in \mathbb{R}_{\geq 0}^{2 \times 2}$. Denote the estimate of $\zeta_{t,i}^1$ as $\hat{\zeta}_{t,i}^1$. Since the drift $\zeta_{t,i}^1$ follows a Random Walk as stated in equation (41), we have $\hat{\zeta}_{t,i}^1 = \zeta_{t-1,i}^1$.

Since the covariance matrix $\mathbf{Q}_t^{\text{drift}}$ is intended to capture variations in the residuals $\mathbf{y}_{t,i}$ under non-stationary conditions, we assume $\mathbf{Q}_t^{\text{drift}} \propto (\mathbf{y}_{t,i} - \mathbf{y}_{t-1,i})(\mathbf{y}_{t,i} - \mathbf{y}_{t-1,i})^T$, where $(\cdot)^T$ is the transpose operation. This formulation directly reflects the instantaneous change in the residual and is therefore well-suited for modeling drift in non-stationary environments. Consistently, the $\tau$-step drift is modeled as $\zeta_{t,i}^\tau = \zeta_{t,i}^1 + (\tau - 1)(\mathbf{y}_{t,i} - \mathbf{y}_{t-1,i})$, which linearly extrapolates future drift based on the most recent residual increment.

Under the above modeling assumptions, the $\tau$-step drift estimate is given by $\hat{\zeta}_{t,i}^\tau = \hat{\zeta}_{t,i}^1 + (\tau - 1)(\mathbf{y}_{t,i} - \mathbf{y}_{t-1,i})$, which extrapolates the drift linearly based on the most recent residual increment. The corresponding covariance estimate is defined as $\hat{\mathbf{Q}}_t^{\text{drift}} = (\mathbf{y}_{t,i} - \mathbf{y}_{t-1,i})(\mathbf{y}_{t,i} - \mathbf{y}_{t-1,i})^T + \gamma \mathbf{I}_{2 \times 2}$, which preserves consistency with the residual-based drift formulation while improving numerical stability. Here, $\mathbf{I}_{2 \times 2}$ denotes the $2 \times 2$ identity matrix, and $0 < \gamma \ll 1$ is a small positive regularization (jitter) constant introduced to ensure that $\hat{\mathbf{Q}}_t^{\text{drift}}$ remains invertible.

Consider $\mathbf{C}_{t,i}^\tau$ as the covariance for $\hat{\mathbf{x}}_{(t+\tau|t),B}^{A_i}$. The onboard algorithm $A_i$'s corrected $\tau$-step look-ahead prediction $\tilde{\mathbf{x}}_{(t+\tau|t),B}^{A_i}$ and the associated covariance $\tilde{\mathbf{C}}_{t,i}^\tau$ are calculated as:

$$\tilde{\mathbf{x}}_{(t+\tau|t),B}^{A_i} = \hat{\mathbf{x}}_{(t+\tau|t),B}^{A_i} - \hat{\zeta}_{t,i}^\tau \tag{42}$$

$$\tilde{\mathbf{C}}_{t,i}^\tau = \mathbf{C}_{t,i}^\tau + \hat{\mathbf{Q}}_t^{\text{drift}} \tag{43}$$

The rest of the SBKCF algorithm is similar to that of KCF, stated as follows:

*Local prediction phase:*

$$(\mathbf{C}_{(t+\tau|t),B}^{L_i})^{-1} = \frac{1}{2}\left((\mathbf{C}_{(t-1+\tau|t-1),B}^i)^{-1} + (\tilde{\mathbf{C}}_{t,i}^\tau)^{-1}\right) \tag{44}$$

$$\hat{\mathbf{x}}_{(t+\tau|t),B}^{L_i} = \left((\mathbf{C}_{(t-1+\tau|t-1),B}^i)^{-1} + (\tilde{\mathbf{C}}_{t,i}^\tau)^{-1}\right)^{-1}\left((\tilde{\mathbf{C}}_{t,i}^\tau)^{-1}\tilde{\mathbf{x}}_{(t+\tau|t),B}^{A_i} + (\mathbf{C}_{(t-1+\tau|t-1),B}^i)^{-1}\hat{\mathbf{x}}_{(t-1+\tau|t-1),B}^i\right) \tag{45}$$

*Communication phase:* The $i^{th}$ robot sends the information $\{t, i, \hat{\mathbf{x}}_{(t+\tau|t),B}^{L_i}, \mathbf{C}_{(t+\tau|t),B}^{L_i}\}$ and receives the information $\{t, j, \hat{\mathbf{x}}_{(t+\tau|t),B}^{L_j}, \mathbf{C}_{(t+\tau|t),B}^{L_j}\}$ from its communicating neighbors $j \in \Omega_{t,i}$.

*Social prediction phase:* With $n_{t,i} = |\Omega_{t,i}|$, we have

$$(\mathbf{C}_{(t+\tau|t),B}^i)^{-1} = \frac{1}{n_{t,i}+1}\sum_{\forall j \in \Lambda_{t,i}}(\mathbf{C}_{(t+\tau|t),B}^{L_j})^{-1} \tag{46}$$

$$\hat{\mathbf{x}}_{(t+\tau|t),B}^i = \left(\sum_{\forall j \in \Lambda_{t,i}}(\mathbf{C}_{(t+\tau|t),B}^{L_j})^{-1}\right)^{-1}\sum_{\forall j \in \Lambda_{t,i}}(\mathbf{C}_{(t+\tau|t),B}^{L_j})^{-1}\hat{\mathbf{x}}_{(t+\tau|t),B}^{L_j} \tag{47}$$

### A.3.3 Parametric Study

Table 1 shows the optimal learning rates for which the mean of the average social cumulative prediction loss of all the robots at $t = 140$ *sec.* (i.e., $\frac{1}{6}\sum_{i=1}^6 L_{1400,i}^S$) is minimum for different $T_o$ values. From table 1, the best mean performance occurs at $T_o = 45$, with $\eta_w = 8.8$ and $\eta_\alpha = 3.2$. A similar study is performed to get the optimal learning rates for which the standard deviation of the average social cumulative prediction loss of all the robots at $t = 140$ *sec.* (i.e., $\frac{1}{6}\sum_{i=1}^6 L_{1400,i}^S$) is minimum for different $T_o$ values, which shows that the standard deviation is the least at around $T_o = 25$, with $\eta_w = 10.0$ and $\eta_\alpha = 5.4$. Figures 4a and 4b show the contour plots for the mean and standard deviation of the average social cumulative prediction loss of all the robots at $t = 140$ *sec.* (i.e., $\frac{1}{6}\sum_{i=1}^6 L_{1400,i}^S$) versus the combination of learning parameters $(\eta_\alpha, \eta_w)$ for $T_o = 45$ and $T_o = 25$, respectively. Note the sensitivity in both the mean and standard deviation due to the learning parameter $\eta_\alpha$.

### A.3.4 DOME Weights' Behavior

A screenshot of a sample simulation run is shown in Fig. 2b. The dynamic behavior of DOME weights $w_{3j}(t)$, $j \in \Lambda_{t,3}$ for a time duration of 5 seconds is shown in Fig. 5 along with the corresponding cumulative local prediction losses for the $3^{rd}$ robot and its nominal neighbors robot 2 and robot 4 (note that random link drops occur, due to which the nominal neighbors may not be neighbors at all times), along with the cumulative social prediction loss for the $3^{rd}$ robot.

### A.3.5 DOME Ablation Study

To identify which components of the DOME fusion algorithm contribute most to the performance gains, an ablation study is conducted: see Fig. 6. Without the periodic reset mechanism, the DOME algorithm's average cumulative 10-step look-ahead prediction loss (median) increases by 1025.6%, with a corresponding substantial rise in loss variance. This result highlights the periodic reset mechanism as a critical component of DOME.

Bypassing the DOME algorithm's local prediction fusion phase (i.e., $\alpha_i(t) = 1$) leads to an increase in the average cumulative 10-step look-ahead prediction loss (median) by 1014.4%, whereas using fixed (equal) weighting in the local prediction fusion phase (i.e., $\alpha_i(t) = 0.5$) leads to an increase by 449.51%. This indicates that both the local fusion layer and its adaptive weighting are important to the performance of the DOME algorithm, with the impact of adaptive weighting being approximately half that of the local fusion layer.

Bypassing the DOME algorithm's social prediction fusion phase (i.e., $w_{ii}(t) = 1$) leads to an increase in the average cumulative 10-step look-ahead prediction loss (median) by 1392.95%, whereas using equal weighting in the local prediction fusion phase (i.e., equal $w_{ij}(t)$) leads to an increase by 183.52%. This indicates that the social fusion layer has a greater impact on the DOME algorithm's performance than its adaptive weighting.

Using equal weighting in both the local and social prediction fusion phases (i.e., ACF: $\alpha_i(t) = 0.5$ and equal $w_{ij}(t)$) leads to an increase in the average cumulative 10-step look-ahead prediction loss (median) by 1232.75%. This demonstrates that adaptive weighting across both the local and social fusion layers is a key contributor to the DOME algorithm's performance gains.

Taken together, the results suggest a nuanced hierarchy of importance: adaptive weighting is more influential in the local fusion layer than in the social layer, while the social fusion layer itself has a greater overall impact on performance than the local fusion layer. The periodic reset mechanism, however, remains a critical component overall.

### A.3.6 Low versus High Communication Link Drop Probability

DOME is evaluated under varying communication link drop probabilities $p_{ld}$; see Fig. 7. The results indicate that as $p_{ld}$ increases from 0.01 to 0.5, the average cumulative 10-step look-ahead prediction loss (median) rises by 80.85%, yet remains within the 100-loss range (median).

### A.3.7 Low versus High Prediction Bias/Drift Rate

DOME is evaluated under varying bias or drift rates $c_{t,i}$ for inaccurate onboard prediction algorithms $A_i$; see Fig. 8. The results indicate that as $c_{t,i}$ increases from 0.01 to 10, the average cumulative 10-step look-ahead prediction loss (median) rises by 705.22%, yet remains within the 100-loss range (median).

### A.4 Full Observability Assumption's Validity in Practice

The full observability assumption in DOME is practically motivated by two operational paradigms in robotic swarms (as described in Section 3):

1. Decentralized Sensing and Learning (costly): Robots possess high-grade (negligible noise), long-range, wide–field-of-view sensors capable of independently tracking the target.

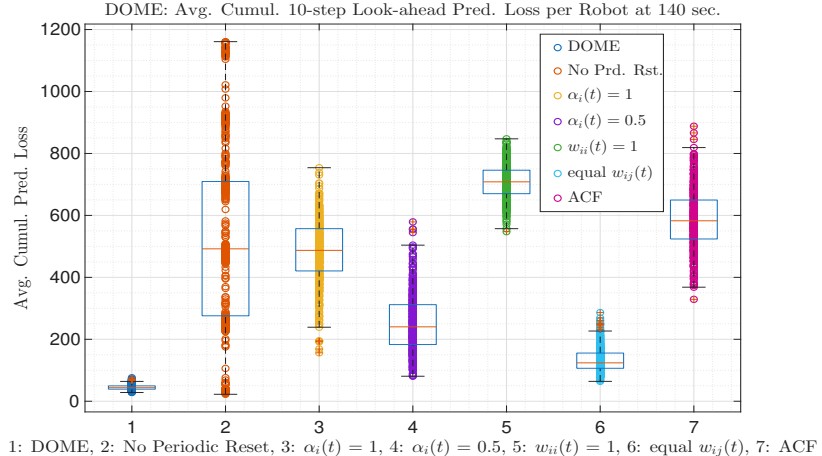

Figure 6: Ablation study of DOME illustrating the contribution of individual components to overall performance. (1) Full DOME; (2) DOME without periodic reset; (3) DOME with $\alpha_i(t) = 1$; (4) DOME with $\alpha_i(t) = 0.5$; (5) DOME with $w_{ii}(t) = 1$; (6) DOME with equal $w_{ij}(t)$; (7) ACF: DOME with $\alpha_i(t) = 0.5$ and equal $w_{ij}(t)$.

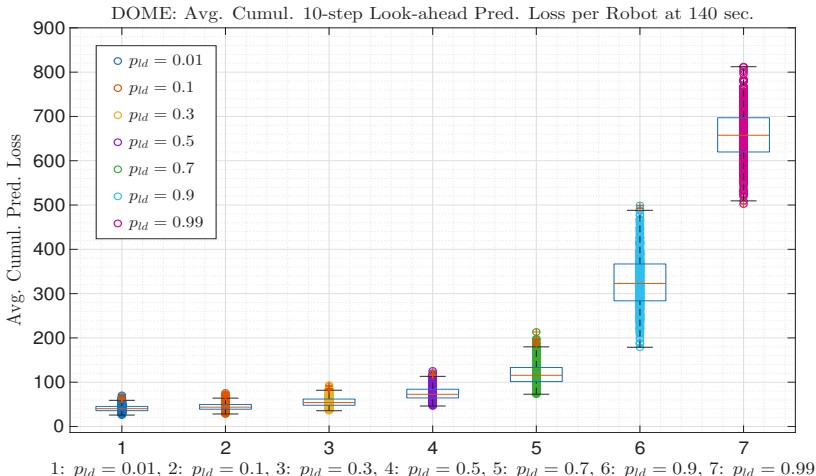

Figure 7: Performance of DOME under varying communication link drop probabilities $p_{ld}$, ranging from low ($p_{ld} = 0.01$) to high ($p_{ld} = 0.99$).

2. Centralized Sensing with Decentralized Learning (cost-effective and scalable): A subset of "leader" robots or centralized assets (e.g., ground radar, AWACS, or satellite feeds) possess high-grade (negligible noise), long-range, wide–field-of-view sensors. These assets broadcast accurate target positions to the swarm via one-way communication, providing a "teaching signal" for the decentralized DOME fusion framework.

The second paradigm (Centralized Sensing with Decentralized Learning) is particularly relevant in high-stakes surveillance or disaster-management scenarios, where the swarm is often supported by high-altitude assets like AWACS (Airborne Warning and Control System) or powerful ground-based radar networks. These assets provide a global fix on the target's position: the centralized assets provide the "training" (ground truth), while the decentralized swarm handles the "execution" (predictive-tracking). This setup justifies deploying DOME on simple hardware, as the expensive task of truth generation is offloaded to centralized assets, while robots perform lightweight weight updates. The resulting scalability is a key practical advantage, allowing swarms to grow to hundreds of robots without saturating communication bandwidth.

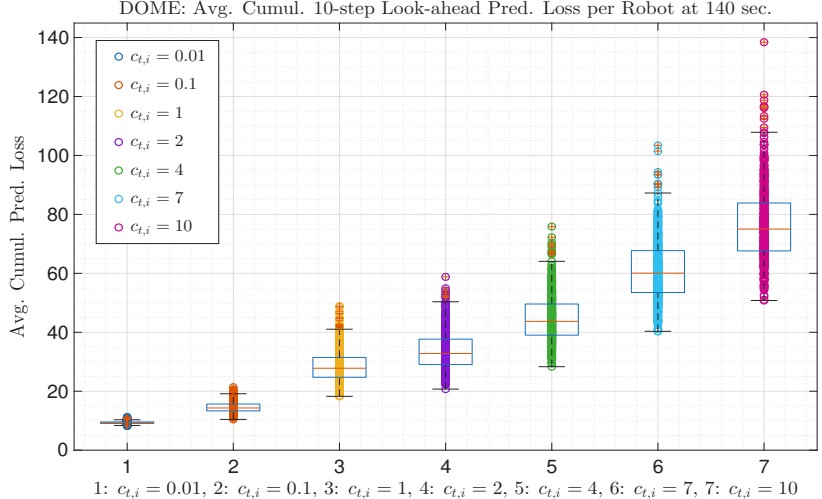

Figure 8: Performance of DOME under varying prediction bias/drift rate $c_{t,i}$ for the inaccurate algorithm $A_i$, ranging from low ($c_{t,i} = 0.01$) to high ($c_{t,i} = 10$).

## A.5 Effect of Noisy/Biased Observations

**Theoretical Analysis:** Let the noisy observations be denoted as:

$$\tilde{\mathbf{x}}_{t,B}^i = \mathbf{x}_{t,B} + \epsilon_{t,i} \tag{48}$$

where $\epsilon_{t,i} \in \mathbb{R}^2$ is a bounded arbitrary noise in the $i^{th}$ robot's observation. Given that the loss function is $l(\mathbf{x},\mathbf{y}) = \min(\|\mathbf{x}-\mathbf{y}\|_2/D_0, 1)$, assume $D_0$ to be sufficiently large such that $l(\mathbf{x},\mathbf{y}) = \|\mathbf{x}-\mathbf{y}\|_2/D_0$. Therefore, the noisy loss $\tilde{l}_{t,i}^L$ is:

$$\tilde{l}_{t,i}^L = \frac{\|\hat{\mathbf{x}}_{t,B}^{L_i} - \tilde{\mathbf{x}}_{t,B}^i\|_2}{D_0} = \frac{\|\hat{\mathbf{x}}_{t,B}^{L_i} - \mathbf{x}_{t,B} - \epsilon_{t,i}\|_2}{D_0} \tag{49}$$

And the ideal loss $l_{t,i}^L$ is:

$$l_{t,i}^L = \frac{\|\hat{\mathbf{x}}_{t,B}^{L_i} - \mathbf{x}_{t,B}\|_2}{D_0} \tag{50}$$

Using the reverse triangle inequality, we get:

$$|\tilde{l}_{t,i}^L - l_{t,i}^L| \le \frac{\|\epsilon_{t,i}\|_2}{D_0} \tag{51}$$

From the above, we have:

$$l_{t,i}^L - \frac{\|\epsilon_{t,i}\|_2}{D_0} \le \tilde{l}_{t,i}^L \le l_{t,i}^L + \frac{\|\epsilon_{t,i}\|_2}{D_0} \tag{52}$$

The weight update rule $w_{ij}(t)$ is given by:

$$w_{ij}(t) = \frac{\hat{w}_{jj}(t-1)\exp(-\eta\tilde{l}_{t,j}^L)}{\sum_{j'\in\Lambda_{t,i}}\hat{w}_{j'j'}(t-1)\exp(-\eta\tilde{l}_{t,j'}^L)} \tag{53}$$

Consider the ratio of weights $w_{ij}$ and $w_{ik}$, such that $j,k \in \Lambda_{t,i}$:

$$\frac{w_{ij}(t)}{w_{ik}(t)} = \frac{\hat{w}_{ij}(t-1)}{\hat{w}_{ik}(t-1)} \frac{\exp(-\eta\tilde{l}_{t,j}^L)}{\exp(-\eta\tilde{l}_{t,k}^L)} \tag{54}$$

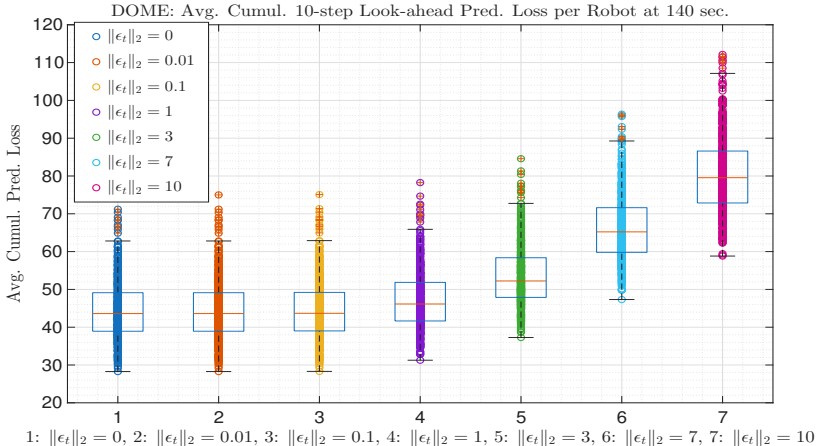

Figure 9: Performance of DOME under varying observation noise or bias magnitude $\|\epsilon_t\|_2$ (in meters), ranging from low ($\|\epsilon_t\|_2 = 0$) to high ($\|\epsilon_t\|_2 = 10$).

By substituting the bounds from equation (52), we establish the inequality:

$$\frac{\hat{w}_{ij}(t-1)\exp(-\eta l_{t,j}^L)\exp(-\eta\frac{\|\epsilon_{t,j}\|_2}{D_0})}{\hat{w}_{ik}(t-1)\exp(-\eta l_{t,k}^L)\exp(\eta\frac{\|\epsilon_{t,k}\|_2}{D_0})} \leq \frac{w_{ij}(t)}{w_{ik}(t)} \leq \frac{\hat{w}_{ij}(t-1)\exp(-\eta l_{t,j}^L)\exp(\eta\frac{\|\epsilon_{t,j}\|_2}{D_0})}{\hat{w}_{ik}(t-1)\exp(-\eta l_{t,k}^L)\exp(-\eta\frac{\|\epsilon_{t,k}\|_2}{D_0})} \tag{55}$$

The final bounded relationship is expressed as:

$$\underbrace{\frac{\hat{w}_{ij}(t-1)\exp(-\eta l_{t,j}^L)}{\hat{w}_{ik}(t-1)\exp(-\eta l_{t,k}^L)}}_{\text{ideal relative weight update ratio}} \cdot \exp\left(-\eta\frac{\|\epsilon_{t,j}\|_2+\|\epsilon_{t,k}\|_2}{D_0}\right)$$
$$\leq \frac{w_{ij}(t)}{w_{ik}(t)} \leq \tag{56}$$
$$\underbrace{\frac{\hat{w}_{ij}(t-1)\exp(-\eta l_{t,j}^L)}{\hat{w}_{ik}(t-1)\exp(-\eta l_{t,k}^L)}}_{\text{ideal relative weight update ratio}} \cdot \exp\left(\eta\frac{\|\epsilon_{t,j}\|_2+\|\epsilon_{t,k}\|_2}{D_0}\right)$$

A similar analysis can be shown for the local weights $\alpha_i(t)$ as well.

**Empirical Analysis:** Consider the operational paradigm of centralized sensing with decentralized learning, i.e., a centralized sensing asset broadcasting the (noisy) observed target position $\tilde{\mathbf{x}}_{t,B} = \mathbf{x}_{t,B} + \epsilon_t$, where $\epsilon_t \in \mathbb{R}^2$ is a constant magnitude noise with randomly changing direction. The DOME algorithm is evaluated under varying observation noise magnitudes $\|\epsilon_t\|_2$ (see Fig. 9). As $\|\epsilon_t\|_2$ increases from 0 to 1 meter, the median average cumulative 10-step look-ahead prediction loss increases by only 5.83%. In contrast, when $\|\epsilon_t\|_2$ increases from 0 to 10 meters, the corresponding loss rises by 82.45%. Yet the loss remains within the 100-loss range (median) for $\|\epsilon_t\|_2 = 10$ meters.

### A.6 Implications of a Fixed Periodic Reset

As noted in Section 4 of the paper, "the DOME weights $w_{ij}(t)$ are analyzed without considering periodic resets to examine their convergence behavior immediately after and just before a reset." Accordingly, the convergence result in Theorem 1 applies within each stationary epoch (i.e., the interval between resets). In practice, if the learning rates are sufficiently large, DOME can identify the optimal predictor before the environment shifts, while the reset mechanism addresses the shifts themselves. If the learning rates are low, the weights may not reach their convergence values before a reset occurs; however, their convergence behavior still follows the analysis in Theorem 1.

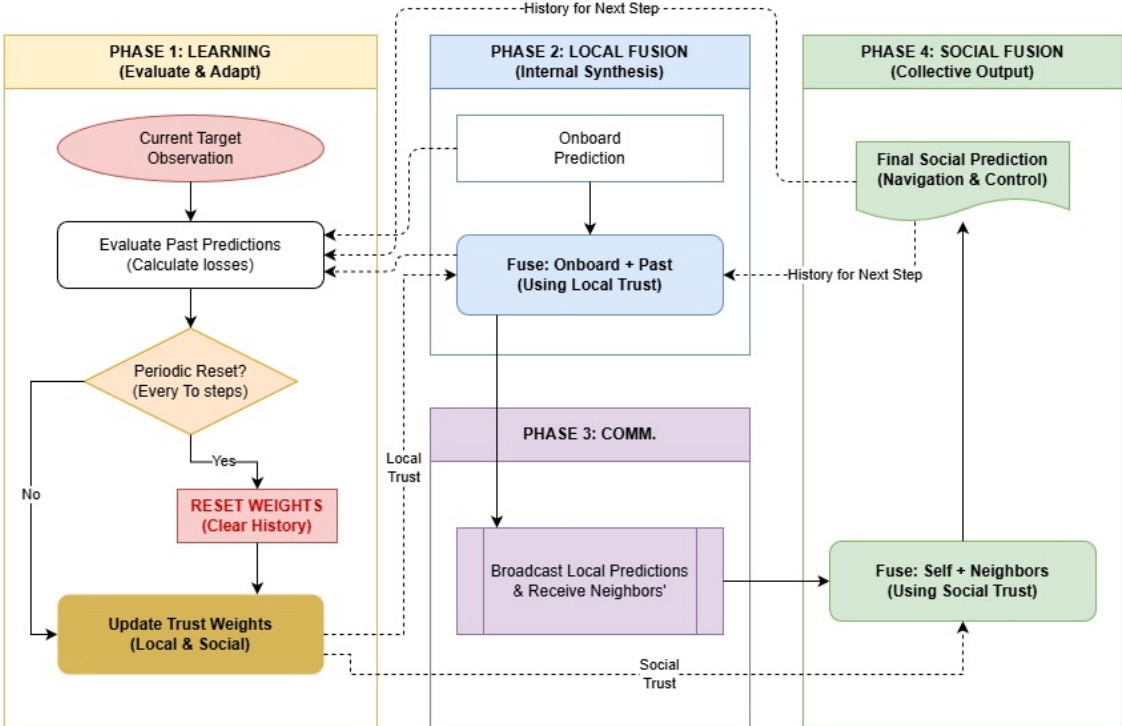

Figure 10: Conceptual Overview of DOME Fusion Framework

**Theoretical Implications of a Fixed $T_o$:** Theoretically, the ideal setting is for the periodic reset interval ($T_o$) to match the environment's switching interval (i.e., the inverse of the bias or environment switching frequency), allowing DOME to converge within each stationary segment. If resets occur too frequently, DOME may remain in a high-variance transient state without fully converging to the best neighbor. Conversely, if resets are too infrequent, DOME can retain large weights on a now-biased neighbor, leading to adaptivity lag.

**Practical Implications of a Fixed $T_o$:** While a fixed $T_o$ might be suboptimal if the frequency of environmental shifts varies drastically, empirical results suggest the algorithm exhibits a robust performance basin rather than a sharp peak, making it resilient to such mismatches. Specifically, the simulation study (Section 6) utilizes an event-based switching bias model (Fig. 2a) where the bias dynamics are inherently stochastic and variable-frequency (switching probabilities $p_{gb} = 0.2, p_{rb} = 0.5$). Despite this variability in the 'true' switching frequency, the parametric study (Appendix A.3.3, Table 1) demonstrates that the DOME algorithm maintains high performance across a broad range of fixed $T_o$ values. As shown in Table 1 (Appendix A.3.3), while the minimum for cumulative prediction loss occurs at $T_o = 45$, the performance degradation for $T_o$ values ranging from 25 to 60 w.r.t. $T_o = 45$ is gradual and relatively minor (loss increases from $\approx 30.5$ to $\approx 34.5$).

