# OpenReview forum: "DOME: Distributed Online Learning based Multi-Estimate Fusion for Cooperative Predictive Target Tracking Using a Robotic Swarm"
_TMLR — Accepted by TMLR_

### Review · Reviewer_fa66 · 2025-12-15

**Summary Of Contributions:**

In this work, the authors consider the problem of a swarm of robots trying to predict the position of a target over time, when each of the robots in the swarm has access to potentially different sensors and a different trajectory computation algorithm, while having to deal with the movement of the robots.
The robots can communicate with their neighbours to share their trajectory updates and evaluate the relative trust they have in their own trajectories versus the trajectories of their neighbours (which might be calculated using more performant sensors/ algorithms).

The algorithm relies on exponential weights (Cesa-Bianchi and Lugosi, 2006), which is a well-studied approach in the non-stochastic learning paradigm. This structure of exponential weights is rather old and has been widely built upon in the literature, including in many applications that could be of interest to expand the current work (referring to the literature on EXP3 and bandits, which studies variants with expert advice (similar to the social structure, handling potentially changing sets of neighbours, delayed observations or faster convergence in easier environments). This being said, the structure of the problem is extremely complex and specific enough that starting with a simple version of exponential weights is probably unavoidable.

In terms of theoretical results, the authors prove that the social weights converge, which means that eventually each robot uses the prediction of the most trustworthy of them or their neighbours, even in challenging situations where the communication between robots might drop, or when the best robot changes over time (due to a problem with a sensor, for example). This robustness and adaptivity are an essential part of the exponent weights methods and could be expanded upon further.

The authors also propose a very detailed experimental setup and benchmark comparison with includes both Covariance-based methods and online learning methods. In that setting, the proposed algorithm achieves much better performance than the baselines. The experiments include validation in a ROS-Gazebo simulated environment.

**Audience:**

Yes

**Audience Explanation:**

I think that the learning theory community, which is a target audience of TMLR,  could be interested in this more applied work, as there is always interest in new ways to bridge the gap between theory and practice. For example,  the early version of exponential weights used in the DOME algorithm could be replaced by a more recent and problem-specific version of the problem (for example, tailored for communication in graphs, or with faster convergence in specifically easy environments).

**Broader Impact Concerns:**

The most straightforward application of the project seems pretty clearly applied to drones, which could be discussed more in the broader impact statement due to the wide range of uses of drones.
The considerations of distributed methods for privacy in the other discussed applications are interesting.

**Claims And Evidence:**

Yes

**Claims Explanation:**

The claims, as stated in the theoretical results, follow detailed proofs. There is, however, a point that needs to be emphasised. In the Online Learning literature, results would typically not be stated in terms of asymptotic convergence but rather with cumulative regret guarantees, which are much stronger.

Regarding the experiments, it appears that all the necessary elements have been provided to reproduce them. Some of the methods in the baseline appear fairly older, so it would be beneficial to have an additional discussion/justification for their use in practice/inexistence of other methods, to confirm that their choice is representative of the current state of the art and that the performance results announced are a fair representation to the state of the art.

**Requested Changes:**

The algorithm has numerous elements that are difficult to keep track of; it would be helpful to have, first, a concise overview of how the algorithm functions in simple terms before introducing and defining all the specifics. Some aspects of the algorithm aren't very clear even after reading the paper: for example, figuring out why the algorithm needs to restart after T_0 steps was difficult, as the couple of lines about it on page 5 are easy to miss.

Some of the variables (for example, J_i) haven't made it to the glossary, so it would be nice to add them for clarity.

As discussed in the previous section, it would be beneficial to argue that the baseline in the experiment represents the state of the art.

---

> ### Author Response · Authors · 2026-02-07
>
> The authors thank the reviewer for highlighting the paper’s contributions and for providing valuable insights that help improve the paper and guide future work.
>
> - The authors would like to emphasize that the loss function is assumed to be bounded but not necessarily convex. As a result, classical regret guarantees typical of the online learning literature are not directly applicable, and the analysis is instead conducted in terms of the convergence of weights under these more general loss assumptions.
>
> - Although some baselines (e.g., Kalman Fusion, Covariance Intersection, and Covariance Union) originate from earlier works, they remain widely used in practice due to their simplicity, low computational complexity, and closed-form implementation. Recent works such as Chang et al. (2021), Daass et al. (2021), Wang et al. (2021), and Jia et al. (2023), cited in the paper's introduction, use these methods for multi-estimate fusion. The proposed DOME algorithm is likewise lightweight and analytically tractable, yet achieves better performance without requiring assumptions regarding consistency and/or correlation.
>
> - The authors will prepare a short paragraph and, if necessary, an overview diagram to explain the algorithm concisely before presenting the specifics. The overview diagram may be moved to the appendix to ensure the main paper remains within the 12-page limit.
>
> - The authors will add a short paragraph clearly explaining the motivation and mechanism behind the periodic reset.
>
> - The authors will include all missing variables in the glossary, including $J_i$.
>
> - The authors will add justification for the use of these baselines as state-of-the-art in practice within a new “Related Work” section.
>
> - **Modified Broader Impact Statement**: The proposed DOME fusion algorithm enhances collaborative prediction accuracy in decentralized multi-agent systems operating under communication uncertainty and prediction biases, with particular relevance to multi-drone systems, where reliable distributed prediction and fusion are critical for applications such as surveillance, environmental monitoring, disaster response, and autonomous navigation. While developed for cooperative predictive target tracking using a robotic swarm, the DOME framework generalizes across domains requiring decentralized, privacy-preserving, and adaptive prediction fusion. Potential applications span finance, where distributed trading models collaboratively forecast market trends; weather and climate forecasting, where regional prediction centers integrate heterogeneous models; urban sensing and environmental monitoring, where sensor networks fuse multi-modal data; and healthcare, where institutions jointly improve predictive diagnostics without sharing sensitive data. In smart grids and supply chains, DOME can enable robust coordination among nodes by cooperatively predicting energy availability and demand under uncertainty.
>
>
> ***The above-mentioned additions and modifications are being incorporated into the revised version of the paper. The authors welcome any further suggestions from the reviewer if additional changes would be helpful.

---

### Review · Reviewer_sQqg · 2026-01-04

**Summary Of Contributions:**

This paper investigates target tracking via a fleet of robots in a cooperative multi-agent setting. Each agent has it's own sensors and individual prediction algorithm of the targets future position. All controllable agents, as well as the target, are represented by 2D position vectors and heading angles in a 2D plane. Notably the controllable agents cannot communicate with every other agent, but only those that have communication links with each other. This particular problem setting has a number of sources of error and uncertainty, including communication failures/drops and high individual prediction biases. The DOME fusion algorithm is proposed to address these challenges. The algorithm explicitly models communication failures, bias accumulation (via fusion weight resets), and more to handle these challenges. Empirical results in simulated settings demonstrate that DOME achieves better performance (measured via accumulated prediction loss over 60 seconds). Moreover, with more robots DOME performance improves, similar to the improvement shown by baselines.

**Audience:**

Yes

**Audience Explanation:**

Learning based methods are used in this paper to update the weights in DOME to improve tracking performance. It is interesting to see how they handle the multi-agent cooperative setting and model an incomplete communication graph setting.

**Claims And Evidence:**

Yes

**Claims Explanation:**

The algorithm looks sound and the empirical results back up the claims in the paper regarding the performance of DOME.

**Requested Changes:**

For a learning based approach with empirical testing, I typically would expect there to be ablation studies on the proposed method. Part of the paper's investigation should include not just the main results, but results that highlight what exactly makes DOME better and the contributions of its various components. Another reasonable ablation would be to test DOME under specific settings to highlight how it best handles the challenges the paper proposes to address like communication failures. For example testing the performance of DOME under a high vs low communication failure rate.

---

> ### Author Response · Authors · 2026-02-07
>
> The authors thank the reviewer for highlighting the paper’s contributions and for providing valuable insights that help improve the paper.
>
> - The authors will include a thorough ablation study to identify which components of the DOME algorithm contribute most to the performance gains.
>
> - The authors will add two additional simulation studies evaluating DOME under varying levels of prediction bias/drift rate (via the parameter $c_{t,i}$) and communication failure rate (via the communication-link drop probability $p_{ld}$).
>
> - These studies will be presented in the appendices to ensure the main paper remains within the 12-page limit.
>
>
> ***The above-mentioned additions are being incorporated into the revised version of the paper. The authors welcome any further suggestions from the reviewer if additional changes would be helpful.

---

### Review · Reviewer_GhzN · 2026-02-04

**Summary Of Contributions:**

The paper introduces DOME, an approach proposed to combat high prediction bias and uncertainty in cooperative distributed online learning system deployed for predictive target tracking. Both theoretical analysis and empirical results were presented, with DOME performing better than the reported baselines in harsh conditions where communication is highly unreliable.

The work includes sufficient baselines for comparison, with results averaged over multiple runs. Furthermore, the algorithm, hyperparameters, and source code are provided to ensure reproducibility.

**Additional Comments:**

N/A

**Audience:**

Yes

**Audience Explanation:**

The paper addresses a problem that is relevant to a sub-community in TMLR as evidenced by previously published papers within the community.

**Broader Impact Concerns:**

A broader impact statement of the proposed method should be discussed. Although the experiments were conducted in a simulation (ROS Gazebo), it is important to reflect on the broader impact of the algorithm in real-world domains.

**Claims And Evidence:**

Yes

**Claims Explanation:**

The claims presented in the paper were addressed via theoretical and empirical evidence.

**Requested Changes:**

Optional changes
1. Consider moving the related works from the introduction section into a separate section. For example, paragraphs 2, 3, and 4 in Section 1 can be moved to a separate section focused on related works.

---

> ### Author Response · Authors · 2026-02-07
>
> The authors thank the reviewer for highlighting the paper’s contributions and for providing valuable insights that help improve the paper.
>
> - The authors will introduce a dedicated “Related Works” section immediately after the Introduction and move the relevant material from paragraphs 2, 3, and 4 of the Introduction into this section for better organization and clarity.
>
> - **Modified Broader Impact Statement**: The proposed DOME fusion algorithm enhances collaborative prediction accuracy in decentralized multi-agent systems operating under communication uncertainty and prediction biases, with particular relevance to multi-drone systems, where reliable distributed prediction and fusion are critical for applications such as surveillance, environmental monitoring, disaster response, and autonomous navigation. While developed for cooperative predictive target tracking using a robotic swarm, the DOME framework generalizes across domains requiring decentralized, privacy-preserving, and adaptive prediction fusion. Potential applications span finance, where distributed trading models collaboratively forecast market trends; weather and climate forecasting, where regional prediction centers integrate heterogeneous models; urban sensing and environmental monitoring, where sensor networks fuse multi-modal data; and healthcare, where institutions jointly improve predictive diagnostics without sharing sensitive data. In smart grids and supply chains, DOME can enable robust coordination among nodes by cooperatively predicting energy availability and demand under uncertainty.
>
>
> ***The above-mentioned modifications are being incorporated into the revised version of the paper. The authors welcome any further suggestions from the reviewer if additional changes would be helpful.

---

### Review · Reviewer_ovsv · 2026-02-08

**Summary Of Contributions:**

The paper introduces DOME, a distributed online-learning-based fusion method, for cooperative predictive target tracking in robotic swarms. DOME uses exponentially weighted updates and adapts to biased or unreliable predictors. Notable benefits are not relying on covariance information and robustness to packet drops. The method scales well with swarm size, as demonstrated in simulations and ROS-Gazebo experiments. Key limitations are strong observability assumption for computing losses and reliance on a heuristic periodic reset not captured by the theory.

**Audience:**

Yes

**Audience Explanation:**

Researchers in multi-robot coordination and online learning would appreciate DOME's learning-based approach to prediction fusion, particularly as an alternative to covariance-based methods under bias. The scalable and communication-robust aspects for cooperative tracking problems could be useful to practitioners.

**Broader Impact Concerns:**

The work has no major ethical concerns, but its applicability to surveillance or monitoring tasks suggests potential dual-use considerations. A brief acknowledgement of responsible deployment in the existing Broader Impact Statement would be appropriate.

**Claims And Evidence:**

Yes

**Claims Explanation:**

Major claims of adapting to biased predictors and packet drops are supported by clear empirical evidences that show improved performance over the chosen baselines. The experimental setup is well documented.

However, assumptions such as full observability and structured bias simplify the problem and may limit real-world generality.

**Requested Changes:**

1. In target tracking, one rarely observes the ground truth position. Explicitly discuss how noisy or biased observations of the target instead of true $x_{t,B}$ would affect the weight updates. Or, clarify the practical implications of assuming access to the true target state. (Critical)

2. The covariance-based methods are mathematically derived under the assumption of zero-mean noise. It is a known that Kalman filters fail under significant unmodeled bias. The problem formulation has a bias state where errors are up to 40m and non-zero mean.
While this highlights DOME's strength, a more rigorous comparison would be an adaptive Kalman Filter or a standard KF with a bias-state augmentation. (Critical)

3. Please add a brief discussion in the main text on sensitivity to mismatch in $T_o$. For example, if the bias dynamics change frequency, does the fixed $T_o$ becomes a problem? (Critical)

4. In (9), the robot computes the social prediction using weighted neighbors. Is this value used only for control or is fed back into the next time step? (8) suggests the previous social prediction is used for local fusion. (Non-critical)

---

> ### Author Response · Authors · 2026-02-11
> **Authors' Response - Part 1**
>
> The authors thank the reviewer for highlighting the paper’s contributions and for providing valuable insights that help improve the paper.
>
> 1. **Full Observability Assumption's Practicality**: The full observability assumption in DOME is practically motivated by two operational paradigms in robotic swarms (as described in the paper's Section 2, paragraph 2):
>
> - Decentralized Sensing and Learning (costly): Robots possess high-grade (negligible noise), long-range, wide–field-of-view sensors capable of independently tracking the target.
>
> - Centralized Sensing with Decentralized Learning (cost-effective and scalable): A subset of "leader" robots or centralized assets (e.g., ground radar, AWACS, or satellite feeds) possess high-grade (negligible noise), long-range, wide–field-of-view sensors. These assets broadcast accurate target positions to the swarm via one-way communication, providing a "teaching signal" for the decentralized DOME fusion framework.
>
> The second paradigm (Centralized Sensing with Decentralized Learning) is particularly relevant in high-stakes surveillance or disaster-management scenarios, where the swarm is often supported by high-altitude assets like AWACS (Airborne Warning and Control System) or powerful ground-based radar networks. These assets provide a global fix on the target's position: the centralized assets provide the "training" (ground truth), while the decentralized swarm handles the "execution" (predictive-tracking). This setup justifies deploying DOME on simple hardware, as the expensive task of truth generation is offloaded to centralized assets, while robots perform lightweight weight updates. The resulting scalability is a key practical advantage, allowing swarms to grow to hundreds of robots without saturating communication bandwidth.
>
>
> 2. **Periodic Reset's Connection with Theorem 1**: As noted in Section 4 (paragraph 2) of the paper, "the DOME weights $w_{ij}(t)$ are analyzed without considering periodic resets to examine their convergence behavior immediately after and just before a reset." Accordingly, the convergence result in Theorem 1 applies within each stationary epoch (i.e., the interval between resets). In practice, if the learning rates are sufficiently large, DOME can identify the optimal predictor before the environment shifts, while the reset mechanism addresses the shifts themselves. If the learning rates are low, the weights may not reach their convergence values before a reset occurs; however, their convergence behavior still follows the analysis in Theorem 1.
>
>
> 3. **Assumptions Regarding the Bias Structure**: The DOME algorithm makes no structural assumptions about bias dynamics. As stated in Section 2 (page 4, Abstract Model for Prediction Algorithm) of the paper, the bias is modeled as arbitrary and unknown, potentially non-Gaussian, and without a known distribution; this arbitrariness also allows for adversarial bias. DOME employs a universal online learning update that does not depend on internal physics models (e.g., random-walk or constant-bias models typically required by Kalman filter variants). The “structured” switching-bias model used in Section 5 (Fig. 2a) served only to generate synthetic data for stress-testing under abrupt failures, and DOME was not given access to this structure.
>
> **The authors will revise the text to clearly distinguish the model-agnostic design of DOME from the structured evaluation scenarios used for validation.
>
> 4. **Practical Implications of Assuming Access to True Target State**: In practice, a standard hierarchical perception architecture can be used, where a high-precision perception module or external asset provides an accurate, current-time reference signal as the ground truth to update the DOME weights. For example, modern LiDAR-Inertial Odometry methods such as LIO-SAM (Shan et al., 2020) and FAST-LIO2 (Xu et al., 2022) achieve centimeter-level accuracy on benchmarks; their estimates can serve as proxy ground truth to compute loss and update fusion weights. Additionally, Yan et al. (2022) show that networked radar resources can provide improved fused target states to distributed agents; such broadcasts can be treated as the true current target position $x_{t,B}$ to update fusion weights.
>
> Shan, T., et al. (2020). "LIO-SAM: Tightly-coupled Lidar Inertial Odometry via Smoothing and Mapping." IEEE/RSJ International Conference on Intelligent Robots and Systems (IROS).
>
> Xu, W., et al. (2022). "FAST-LIO2: Fast Direct LiDAR-Inertial Odometry." IEEE Transactions on Robotics.
>
> Yan, J., et al. (2022). "Radar Sensor Network Resource Allocation for Fused Target Tracking: A Brief Review." Information Fusion.
>
> **The authors will add the above discussion in Section 2 of the revised paper.

---

> ### Author Response · Authors · 2026-02-11
> **Authors' Response - Part 2**
>
> 5. **Effect of Noisy/Biased Observations on Weight Updates**: At the reviewer’s request, the authors conducted a brief mathematical analysis of weight updates under noisy or biased observations, assuming bounded but otherwise arbitrary noise. The analysis shows that the resulting loss remains within a strict interval around the ideal/true loss, which in turn induces a bounded relative distortion in the updated weights.
>
> **The authors will include this analysis in the appendix of the revised paper, along with a simulation study evaluating DOME under varying observation-noise magnitudes.
>
> 6. **Additional Kalman Filter Baseline**: The authors will include a Separate-Bias Kalman Filter (SBKF) (Friedland, 1969; Ignagni, 1990) based baseline in the revised paper. This architecture is preferred over a bias-state augmented Kalman filter because it enables bias estimation without violating the model-agnostic assumption of the problem formulation (i.e., unknown target dynamics). As shown by Friedland (1969), the separate-bias estimator is algebraically equivalent to the state-augmented Kalman filter, decomposing estimation into a bias-free state estimator and a dedicated bias estimator. This structure permits bias estimation from prediction residuals without explicitly modeling target dynamics.
>
> Friedland, B. (1969). Treatment of bias in recursive filtering. IEEE Transactions on Automatic Control, 14(4), 359-367.
>
> Ignagni, M. B. (1990). Separate-bias Kalman estimator with bias state noise. IEEE Transactions on Automatic Control, 35(3), 338-341.
>
> 7. **Theoretical Implications of a Fixed Periodic Reset $T_o$**: Theoretically, the ideal setting is for the periodic reset interval ($T_o$) to match the environment’s switching interval (i.e., the inverse of the bias or environment switching frequency), allowing DOME to converge within each stationary segment. If resets occur too frequently, DOME may remain in a high-variance transient state without fully converging to the best neighbor. Conversely, if resets are too infrequent, DOME can retain large weights on a now-biased neighbor, leading to adaptivity lag.
>
> 8. **Empirical Implications of a Fixed Periodic Reset $T_o$**: While a fixed $T_o$ might be suboptimal if the frequency of environmental shifts varies drastically, empirical results suggest the algorithm exhibits a robust performance basin rather than a sharp peak, making it resilient to such mismatches. Specifically, the simulation study (Section 5) utilizes an event-based switching-bias model (Fig. 2a) where the bias dynamics are inherently stochastic and variable-frequency (switching probabilities $p_{gb} = 0.2, p_{rb} = 0.5$). Despite this variability in the 'true' switching frequency, the parametric study (Appendix A.3.3, Table 1, Fig. 4a and 4b) demonstrates that the DOME algorithm maintains high performance across a broad range of fixed $T_o$ values. As shown in Table 1 (Appendix A.3.3), while the minimum for cumulative prediction loss occurs at $T_o = 45$, the performance degradation for $T_o$ values ranging from $T_o = 25$ to $T_o = 60$ w.r.t. $T_o = 45$ is gradual and relatively minor (loss increases from $\approx 30.55$ to $\approx 34.59$).
>
> 9.  Equation (9) computes the current-time $\tau$-step look-ahead social prediction for any $\tau \geq 1$. Equation (8), in contrast, uses the previous time $\tau$-step look-ahead social prediction for any $\tau \geq 1$. Thus, equation (9) yields the $\tau$-step look-ahead social prediction ($\tau \geq 1$), which is fed into equation (8) at the next time step and is also used for control.
>
> 10. **Modified Broader Impact Statement**:
>
> The proposed DOME fusion algorithm enhances collaborative prediction accuracy in decentralized multi-agent systems operating under communication uncertainty and prediction biases, with particular relevance to multi-drone systems, where reliable distributed prediction and fusion are critical for applications such as surveillance, environmental monitoring, disaster response, and autonomous navigation. For responsible deployment, strict adherence to ethical guidelines is essential when operating autonomous monitoring systems in populated areas.
>
> While developed for cooperative predictive target tracking using a robotic swarm, the DOME framework generalizes across domains requiring decentralized, privacy-preserving, and adaptive prediction fusion. Potential applications span finance, where distributed trading models collaboratively forecast market trends; weather and climate forecasting, where regional prediction centers integrate heterogeneous models; urban sensing and environmental monitoring, where sensor networks fuse multi-modal data; and healthcare, where institutions jointly improve predictive diagnostics without sharing sensitive data. In smart grids and supply chains, DOME can enable robust coordination among nodes by cooperatively predicting energy availability and demand under uncertainty.

---

### Author Response · Authors · 2026-02-19
**Uploaded the paper revision**

The revised version of the paper has been uploaded. The following modifications/additions have been made in the revised version of the paper:

1. A new Related Works section just after the Introduction, with relevant text from the previous Introduction shifted to this new section.

2. Justification for the use of baselines as state-of-the-art in practice in the new Introduction section.

3. A conceptual overview subsection in section 4, giving an overview of DOME, and an overview diagram as Fig. 10 in the Appendix. This overview explains the motivation behind the periodic reset mechanism.

4. Included all the missing variables in the Nomenclature (Appendix A.1).

5. Modified the Broader Impact Statement.

6. Added Ablation Study (check section 6, and Appendix A.3.5, Fig. 6).

7. Added Robustness Study (check section 6), i.e., varying communication failure rate (Appendix A.3.6, Fig. 7), prediction drift rate (Appendix A.3.7, Fig. 8), and noisy observations (Appendix A.5, Fig. 9).

8. Added discussion regarding Sensitivity to Mismatch in $T_o$ (section 6, and Appendix A.6).

9. Added detailed discussion regarding Full Observability Assumption’s Validity in Practice (section 2, and Appendix A.4 ).

10. Added a discussion on Assumptions Regarding the Bias Structure in section 6 as Remark 9.

11. Added a discussion regarding Practical Implications of Assuming Access to True Target State as Remark 4 in section 4.

12. Added a detailed theoretical and empirical analysis of the impact of noisy observations on the DOME weight updates in Appendix A.5.

13. Added a new baseline: Separate Bias Kalman Consensus Fusion (SBKCF), inspired by Separate Bias Kalman Filter (SBKF). Turns out that SBKCF is the best among other baselines. Modified the quantitative results in the abstract, introduction, performance evaluation, and conclusion sections of the paper to reflect this. Added a brief qualitative discussion regarding its performance and modified the plots (Fig. 3a and 3b) in the performance evaluation section. Details regarding SBKCF are added in Appendix A.3.2.

14. The main paper has become 14 pages long. Changed the submission type to `long submission' accordingly.

---

### Decision · Action_Editor_miw5 · 2026-03-19

**Recommendation:** Accept as is

**Audience:**

Yes

**Audience Explanation:**

Researchers in the community of online learning and robotics will find this paper interesting.

**Claims And Evidence:**

Yes

**Claims Explanation:**

All the reviewers agree that the claims made in the submission are well supported by clear empirical evidence.

---

> ### Author Response · Authors · 2026-04-17
>
> The authors thank the Action Editor for managing the review process and recognizing this work. The camera-ready version of the paper has been submitted, along with a link to the video presentation and the code.